# LieWarper: Geometry-Aware Motion Transfer via Lie Algebra

Linsong Shan [1]  Laurence T. Yang [1 2]  Zecan Yang [2 †]  Fukai Guo [3]  Honglu Zhao [1]  Yixuan Geng [1]

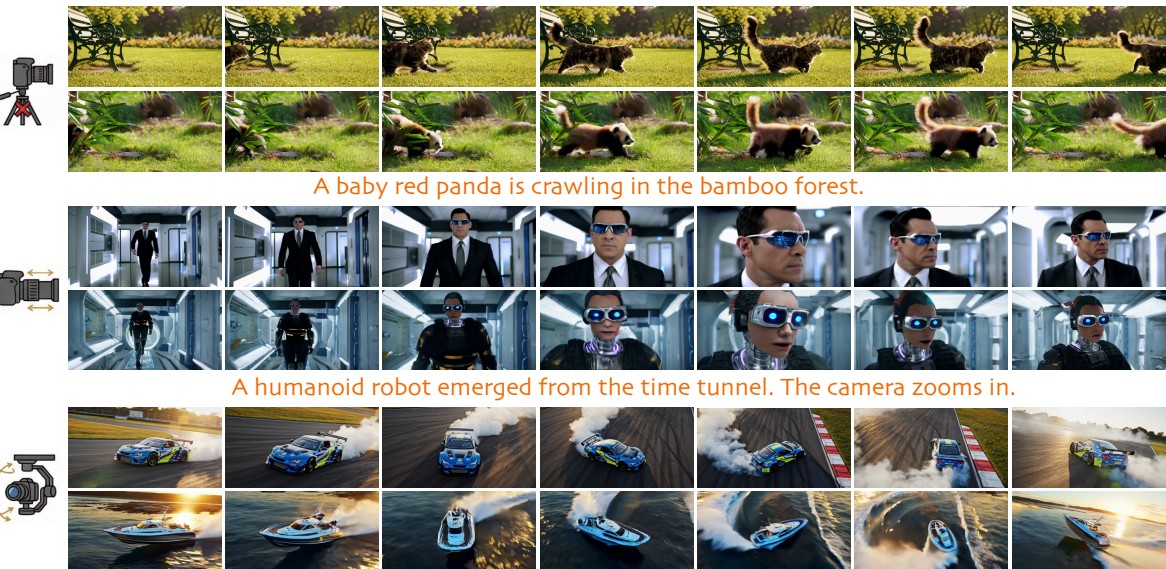

*Figure 1.* **Diverse motion transfer examples.** Demonstrating LieWarper's ability to achieve geometrically consistent, high-fidelity motion transfer across varying complexities.

## Abstract

Video motion transfer aims to synthesize novel content videos that strictly follow the motion trajectories of a reference video. However, existing methods typically operate in Euclidean space, treating motion as unconstrained pixel displacements or linear phase shifts. This simplification frequently causes severe shearing artifacts and perspective collapse under complex camera and object motions. In this work, we present LieWarper, a geometry-aware motion transfer framework that reconceptualizes motion as coordinate evolution on a manifold rather than mere pixel displacement. Specifically, we derive an analytic solver on the $\text{Sim}(2)$ manifold to extract global evolution parameters from noisy optical flow. We then introduce a flow-guided phase modulation mechanism, enabling non-rigid dynamics to undergo coordinate transformation along the evolution path. This approach achieves accurate trajectory transfer while maintaining global geometric integrity. Extensive experiments show that LieWarper significantly outperforms state-of-the-art training-free baselines in both motion fidelity and geometric stability, while maintaining high generation quality. The code is available at GitHub.

[†]Corresponding author. [1]School of Computer Science and Technology, Huazhong University of Science and Technology [2]School of Computer Science and Artificial Intelligence, Zhengzhou University [3]School of Mechanical Engineering and Electronic Information, China University of Geosciences. Correspondence to: Zecan Yang <zecanyang@gmail.com>.

*Proceedings of the $43^{rd}$ International Conference on Machine Learning*, Seoul, South Korea. PMLR 306, 2026. Copyright 2026 by the author(s).

## 1. Introduction

Recent advancements in generative video models have shifted the research focus from merely improving visual fidelity to more fine-grained controllable generation (Peebles & Xie, 2023; Blattmann et al., 2023; Ho et al., 2022). Among various controllable tasks, Video Motion Transfer holds central research value, aiming to synthesize sequences with novel content that strictly adhere to the motion trajectories of a reference video (Wang et al., 2024; Xing et al.,

2024; Yin et al., 2023). From a physical-geometric perspective, video motion is essentially the projection of foreground object dynamics within a transformed camera coordinate system, rather than simple pixel displacement. Therefore, the core challenge of motion transfer is not merely pixel-level alignment but constructing a globally consistent reference frame and, upon this basis, achieving the physical adaptation of source motion and target geometry.

Existing motion transfer paradigms, primarily based on feature warping or attention injection (Wu et al., 2023; Khachatryan et al., 2023; Hu, 2024), have made strides but often fall into a "coordinate blindness." These methods typically adopt an Eulerian perspective, modeling motion simply as unconstrained pixel displacement on a fixed grid. This assumption overlooks the manifold structure of motion, leading to the entanglement of rigid camera motion and non-rigid object deformation. As shown in Figure 2, when confronting complex compound motion (e.g., large-scale camera movement accompanied by object deformation), this lack of geometric constraints often leads to non-physical shearing deformation or volumetric collapse.

To address these challenges, we propose LieWarper, a video motion transfer framework driven by Lie group geometry. Unlike simulating motion in unconstrained Euclidean space, LieWarper models spatiotemporal motion as coordinate evolution on the Sim(2) Lie group manifold. Our core insight is that robust motion transfer requires decoupling video generation into the evolution of a global rigid reference frame and the adaptive projection of local geometric residuals.

- **Analytic Sim(2) Modeling:** We utilize the HOSVD-filtered optical flow to analytically estimate a global similarity transformation, providing a robust rigid transformation reference frame for video generation.

- **Flow-Guided Phase Warping:** We propose a flow-guided phase modulation mechanism that integrates local residuals into the global manifold evolution. This enables local non-rigid dynamics to undergo coordinate transformation along the path of manifold evolution.

- **Joint Optimization:** We leverage the low-degree-of-freedom nature of Lie algebra as a "geometric bottleneck" for joint optimization. With global motion evolving in the manifold tangent space while allowing the residual field to undergo controlled non-rigid adaptation, we preserve the global geometric structure while precisely replicating motion trajectories.

## 2. Related Work

In this section, we review the development of video motion transfer techniques and discuss the relevance of Manifold Geometry for our proposed framework.

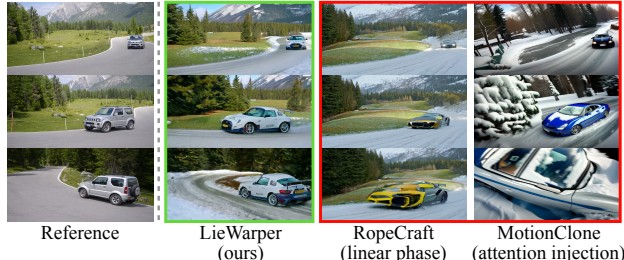

Reference    LieWarper    RopeCraft    MotionClone
            (ours)     (linear phase)   (attention injection)

*Figure 2.* **Global perspective preservation.** Baselines like RoPECraft (linear phase) and MotionClone (attention injection) exhibit **perspective collapse** under complex motion. Our method applies Sim(2) constraints to achieve consistent motion transfer.

### 2.1. Video Motion Transfer

Early approaches relied on explicit structural priors, such as depth maps or edges, to constrain generation (Zhang et al., 2023; Zhang et al.; Wang et al., 2023). While offering spatial control, their reliance on auxiliary annotations restricts applicability in zero-shot scenarios. Moreover, the generation quality is strictly bounded by the accuracy of these off-the-shelf estimators, where any estimation noise inevitably propagates to the synthesized video, causing temporal inconsistencies.

To enhance flexibility, recent research shifts towards modulating internal representations. Implicit methods, including MotionClone (Ling et al., 2024), MotionMaster (Hu et al., 2024), and ConMo (Gao et al., 2025), transfer dynamics by manipulating temporal attention maps, while Wan-Move (Chu et al., 2025) employs specialized motion modules. These approaches effectively capture semantic motion trends but primarily operate in feature space. Since attention mechanisms correlate pixels based on semantic similarity rather than physical coordinates, they often struggle to decouple rigid background shifts from non-rigid object deformations. Consequently, ensuring strict perspective consistency during large-scale camera ego-motion remains challenging without explicit geometric constraints.

Alternatively, explicit alignment strategies aim for precise trajectory control. Methods like MotionCtrl (Wang et al., 2024), CameraCtrl (He et al., 2024) and I2VControl (Feng et al., 2025) incorporate camera pose encoders, whereas training-free approaches like Go-with-the-Flow (Burgert et al., 2025) and RoPECraft (Gokmen et al., 2025) intervene in noise distributions or positional embeddings. While achieving higher precision, these methods typically model motion parameters within Euclidean space. This linear approximation often overlooks the manifold structure of complex perspective transformations, potentially leading to geometric inconsistencies in compound motion scenarios.

## 2.2. Manifold Geometry in Generative Modeling

In classical computer vision, modeling camera ego-motion and perspective projection relies heavily on rigid transformation theories. Mathematically, these continuous transformations are formalized as Lie groups, where SE(3) and Sim(2) provide the algebraic basis for parameterizing 3D rigid body motion and planar similarity, respectively.

Integrating these geometric structures into neural frameworks represents an emerging direction in generative modeling (Bronstein et al., 2021; Hutchinson et al., 2021). Recent advancements have successfully applied Lie group diffusion to 3D structural data and complex geometric manifolds (Yim et al., 2023; Lipman et al., 2022; Li et al., 2025), utilizing Riemannian score matching (De Bortoli et al., 2022; Song et al., 2020) to ensure topological validity.

However, in the video generation, such explicit geometric modeling remains underexplored. Most video diffusion models still rely on data-driven priors rather than mathematical manifold constraints. A fundamental challenge lies in effectively coupling global rigid constraints with the non-rigid dynamics of local objects. This study stands at this intersection, utilizing Lie algebra to establish a reference frame, thereby achieving motion transfer while providing a robust geometric prior for perspective consistency.

## 3. Preliminaries

This section outlines the preliminaries of Latent Diffusion Transformers and establishes the mathematical framework of Sim(2) Lie group geometry, serving as the theoretical foundation for our geometric motion control framework.

### 3.1. Latent Diffusion Transformer & Rotary Positional Encoding (Latent DiT & RoPE)

Modern video generation models (e.g. Sora (Brooks et al., 2024), HunyuanVideo (Kong et al., 2024), Wan (Wan et al., 2025)) widely adopt the DiT architecture. Since the Transformer architecture is inherently *Permutation Invariant*, the model's perception of spatiotemporal structure relies entirely on positional encodings.

Mainstream models employ 3D Rotary Positional Embeddings (3D-RoPE) to model spatiotemporal dependencies (Su et al., 2024). Given spatiotemporal indices $\mathbf{p} = (t, h, w)$ in the latent space, RoPE injects absolute positional information by rotating Query and Key vectors in the complex domain. In standard implementations, to reduce computational complexity, spatial dimensions are typically decoupled; that is, the positional encoding is decomposed into a superposition of independent RoPE($h$) and RoPE($w$).

This implies a strong assumption that motion within the video is mutually independent along the coordinate axes.

However, real-world camera motion (e.g., rotation) introduces strong coupling between height and width. Furthermore, diffusion models rely on the latent variable $\mathbf{z}$ (Rombach et al., 2022) following an i.i.d. Gaussian prior. Unlike warping-based methods that directly translate feature maps—often disrupting the feature statistics without additional regularization, manipulating the positional indices $\mathbf{p}$ offers a non-invasive means to introduce rigid motion while preserving the generative prior of the latent space.

### 3.2. Sim(2) Group and Lie Algebra $\mathfrak{sim}(2)$

As video generation operates on 2D latent feature maps rather than explicit 3D representations, we adopt the Similarity Group (Sim(2)) to model the **projected camera motion** on the imaging plane (Ma et al., 2012). Since the projection of 3D camera dynamics (e.g., dolly, pan, roll) is dominated by similarity transformations, Sim(2) provides a robust geometric approximation without the fragility of ill-posed depth estimation. Unlike the Euclidean Group SE(2) which is limited to rigid rotation and translation, Sim(2) incorporates isotropic scaling, allowing robust parameterization of camera zooming, panning, and in-plane rotation.

**Similarity Group:** The Sim(2) group is a smooth manifold, whose elements $\mathbf{M}$ can be represented as $3 \times 3$ matrices acting on homogeneous coordinates $\mathbf{u} = [x, y, 1]^T$:

$$\mathbf{M} = \begin{bmatrix} s\cos\theta & -s\sin\theta & t_x \\ s\sin\theta & s\cos\theta & t_y \\ 0 & 0 & 1 \end{bmatrix} \in \text{Sim}(2) \qquad (1)$$

where $s \in \mathbb{R}^+$ is the scaling factor, $\theta \in [0, 2\pi)$ is the rotation angle and $(t_x, t_y)$ represents the translation vector.

**Lie Algebra:** To enable optimization, we map the group manifold to its tangent space, the Lie algebra $\mathfrak{sim}(2)$. An element $\boldsymbol{\xi}$ is a four-dimensional vector $\boldsymbol{\xi} = [\sigma, \omega, v_x, v_y]^T$, corresponding to logarithmic scaling ($\sigma = \ln s$), angular velocity ($\omega$), and linear velocity ($v_x, v_y$). The Lie algebra relates to the group through the Exponential Map (Murray et al., 2017). For parameter $\boldsymbol{\xi}$, its induced transformation is:

$$\mathbf{M} = \exp(\boldsymbol{\xi}) = \sum_{k=0}^{\infty} \frac{(\boldsymbol{\xi}^{\wedge})^k}{k!}, \qquad (2)$$

where the $(\cdot)^{\wedge}$operator maps the vector to the corresponding matrix form. The geometric properties of the exponential map guarantee that the generated transformation always lies on the group manifold. This implies that regardless of parameter variation, the induced motion strictly adheres to the manifold structure, mathematically preventing non-physical structural collapse.

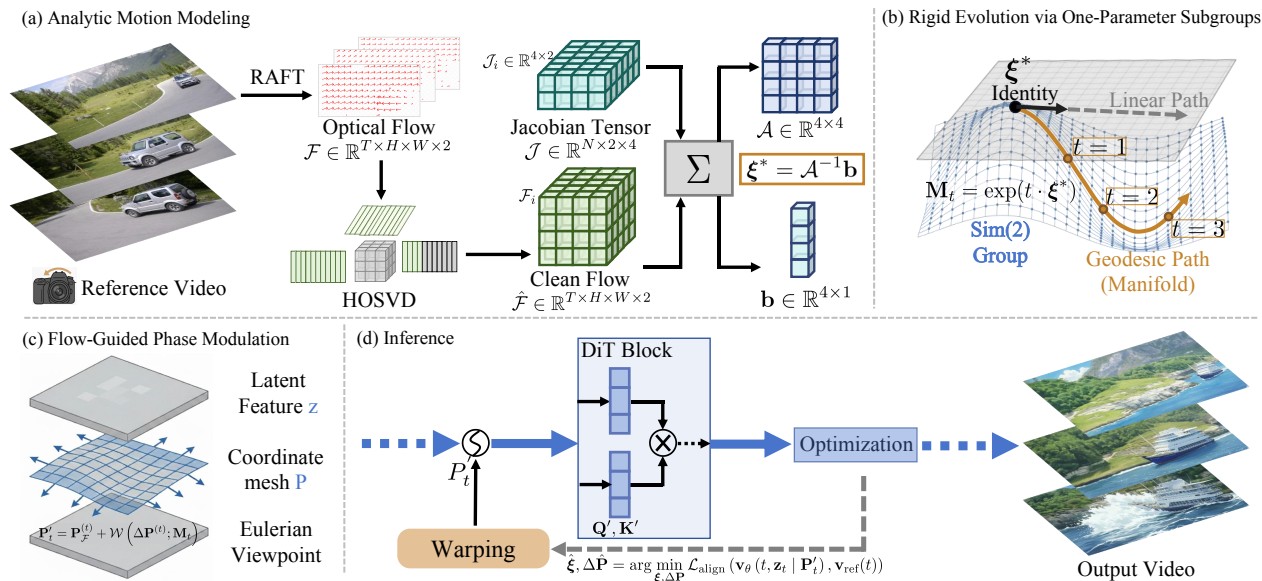

*Figure 3.* **LieWarper Pipeline.** (a) Analytic global motion solution via tensor contraction. (b) Geodesic evolution on Sim(2) for geometric consistency. (c) Feature modulation via Eulerian phase warping. (d) Inference-time joint optimization for motion alignment.

# 4. Method

## 4.1. Framework Overview

Video motion is not merely pixel displacement in pixel space; rather, local dynamics intrinsically evolve within global coordinate transformations. Based on this core insight, we propose the **LieWarper** framework. This framework is designed to establish a Global Rigid Reference Frame to align with camera coordinate transformations, and upon this foundation, introduces Flow-Guided Phase Modulation to achieve the adaptive transport of non-rigid local dynamics over the rigid coordinate evolution.

## 4.2. Analytic Modeling of the Motion Manifold

This section addresses the core geometric consistency problem: how to reconstruct strict global rigid perspective constraints from unstructured, noisy pixel-level observations.

### 4.2.1. MOTION MODELING ON THE MANIFOLD

The dense optical flow field $\mathcal{F} \in \mathbb{R}^{T \times H \times W \times 2}$ describes pixel displacements on the Euclidean plane (Teed & Deng, 2020). However, raw flow is an *ill-posed* representation for control, fraught with occlusion noise and dynamic interference. Direct manipulation of such unstructured data leads to structural tearing. We posit that while local dynamics are non-rigid, they must be spatially transported by the coordinate evolution dictated by the global camera motion. We adopt Sim(2) Lie group manifold to model the coordinate evolution, rather than SE(3) because the lack of camera intrinsics and inevitable depth estimation errors make explicit 3D warping unstable (see Appendix C).

In the tangent space at the identity of Sim(2) (i.e., the Lie algebra $\mathfrak{sim}(2)$), the velocity field is linearly spanned by geometric bases. For any pixel $i$ at coordinates $\mathbf{x}_i$, assuming it follows a rigid motion $\boldsymbol{\xi} \in \mathfrak{sim}(2)$, its theoretical velocity $\mathbf{v}_i$ is governed by:

$$\mathbf{v}_i = \mathbf{J}(\mathbf{x}_i) \cdot \boldsymbol{\xi} = \begin{bmatrix} x_i & -y_i & 1 & 0 \\ y_i & x_i & 0 & 1 \end{bmatrix} \begin{bmatrix} \sigma \\ \omega \\ v_x \\ v_y \end{bmatrix}, \quad (3)$$

where $\mathbf{J}(\mathbf{x}_i) \in \mathbb{R}^{2 \times 4}$ is the **Geometric Jacobian**, mapping abstract Lie parameters $\boldsymbol{\xi}$ to pixel velocity.

### 4.2.2. ROBUST GLOBAL ANALYTIC SOLVER

To mitigate the influence of temporal noise such as video jitter and motion occlusion, we employ HOSVD to process the flow tensor. By performing spectral truncation on the temporal mode, we retain the top-$k$ tensor bases dominating the video's principal motion modes. This operation suppresses incoherent temporal jitter and occlusion artifacts, yielding a cleaner, temporally stabilized flow field $\hat{\mathcal{F}}$, which serves as a reliable basis for fitting the Sim(2) parameters.

The estimation of global motion parameters is then modeled as a least-squares optimization, with the aim of finding the optimal Lie algebra $\boldsymbol{\xi}^*$ that minimizes the distance between the induced flow and $\hat{\mathcal{F}}$. The energy function is:

$$E(\boldsymbol{\xi}) = \sum_{i \in \Omega} \|\mathbf{J}_i \boldsymbol{\xi} - \hat{\mathcal{F}}_i\|_2^2. \quad (4)$$

Since $E(\boldsymbol{\xi})$ is strictly convex, its global minimum is ob-

tained via the normal equations $\nabla_{\boldsymbol{\xi}} E = 0$:

$$\left(\sum_{i \in \Omega} \mathbf{J}_i^T \mathbf{J}_i\right) \boldsymbol{\xi}^* = \sum_{i \in \Omega} \mathbf{J}_i^T \hat{\mathcal{F}}_i. \qquad (5)$$

We define the **Global Structure Tensor** $\mathcal{A} \in \mathbb{R}^{4 \times 4}$ as the left-hand term:

$$\mathcal{A} \triangleq \sum_{i \in \Omega} \mathbf{J}_i^T \mathbf{J}_i. \qquad (6)$$

Physically, $\mathcal{A}$ describes the sensitivity and constraint capability of the image content regarding different rigid transformation modes (scaling, rotation, translation). This contraction operation along spatial dimensions drastically compresses data dimensionality. Correspondingly, the right-hand side is defined as the projection vector $\mathbf{b} \in \mathbb{R}^4$, representing the projection components of the optical flow field onto the Lie algebra basis:

$$\mathbf{b} \triangleq \sum_{i \in \Omega} \mathbf{J}_i^T \hat{\mathcal{F}}_i. \qquad (7)$$

The analytic solution is given by $\boldsymbol{\xi}^* = \mathcal{A}^{-1} \mathbf{b}$. As illustrated in Figure 3(a), this solver acts as a physics-based global filter: HOSVD eliminates jitter in the temporal domain, while tensor contraction cancels out Gaussian noise in the spatial domain via statistical averaging, ensuring $\boldsymbol{\xi}^*$ maintains high geometric consistency even under low signal-to-noise ratios.

### 4.3. Manifold Phase Evolution and Flow-Guided Modulation

Upon obtaining the global Lie algebra parameters $\boldsymbol{\xi}^*$, we propose a manifold phase evolution mechanism. This fuses the global rigid skeleton with local dynamics while respecting the algebraic structure of the transformation group.

#### 4.3.1. RIGID EVOLUTION ALONG ONE-PARAMETER SUBGROUPS

Previous motion control methods typically employ linear interpolation in Euclidean space ($\mathbf{p}_t = \mathbf{p}_0 + t \cdot \mathbf{v}$). However, such additive operations serve only as local approximations. For non-commutative transformations like rotation, linear interpolation generally fails to preserve the group structure (e.g., destroying the orthogonality of rotation matrices), leading to accumulated geometric errors such as "shearing deformation" or "volumetric collapse" over long sequences.

To guarantee the conservation of global motion relationships, we model the camera's trajectory as a One-Parameter Subgroup on the $\mathrm{Sim}(2)$. According to Lie group theory, the Lie algebra $\boldsymbol{\xi}^*$ defines a left-invariant vector field on the manifold. The solved Lie algebra element $\boldsymbol{\xi}^*$ generates a continuous path on the manifold via the exponential map:

$$\mathbf{M}_t = \exp(t \cdot \boldsymbol{\xi}^*) = \sum_{k=0}^{\infty} \frac{(t \cdot \boldsymbol{\xi}^*)^{\wedge k}}{k!} \in \mathrm{Sim}(2) \qquad (8)$$

Unlike linear interpolation, this algebraic evolution ensures that for any time step $t$, $\mathbf{M}_t$ resides on the $\mathrm{Sim}(2)$ manifold (Figure 3 (b)). This ensures the coordinate transformation maintains planar rigidity, preserving orthogonality and isotropic scaling, thus preventing non-physical artifacts common in baselines.

#### 4.3.2. FLOW-GUIDED PHASE WARPING

With the global skeleton $\mathbf{M}_t$ established, the challenge lies in coupling this global reference frame with non-rigid object dynamics. We propose a **Flow-Guided Phase Warping** mechanism to fuse global constraints with local guidance.

To replicate reference trajectories within correct perspective relations, we decouple the target coordinate field into an Observation Prior Field $\mathbf{P}_{\mathcal{F}}^{(t)}$ and a Learnable Geometric Residual Field $\Delta \mathbf{P}^{(t)}$. $\mathbf{P}_{\mathcal{F}}^{(t)}$ is derived via temporal integration of the reference optical flow, serving as a coarse geometric anchor. To introduce controllable fine-grained calibration, we define zero-initialization $\Delta \mathbf{P}^{(t)}$ and transform it via the constructed global rigid skeleton $\mathbf{M}_t$.

Specifically, we mandate that all geometric corrections must evolve within the camera coordinate system rather than diverging freely in pixel space. The core intent of this design is to leverage the analytically solved pure rigid transformation $\mathbf{M}_t$ to perform manifold transport of the learnable geometric information. Consequently, the final coordinate field $\mathbf{P}'_t$ is modeled as:

$$\mathbf{P}'_t = \mathbf{P}_{\mathcal{F}}^{(t)} + \mathcal{W}\left(\Delta \mathbf{P}^{(t)}; \mathbf{M}_t\right). \qquad (9)$$

Here, $\mathcal{W}$ denotes the spatial warping operator driven by $\mathbf{M}_t$, which performs a rigid coordinate transformation on the residual field. Physically, this implies projecting the residual field into the perspective of the current frame, ensuring that any injected detail is geometrically aligned before being superimposed onto the anchor.

The manifold-rectified $\mathbf{P}'_t$ is subsequently injected into the RoPE module. Through rotation in the complex domain, it modulates the Query and Key vectors in Self-Attention:

$$\hat{\mathbf{Q}} = \mathbf{Q} \odot e^{i\langle \boldsymbol{\Theta}, \mathbf{P}'_t \rangle}, \quad \hat{\mathbf{K}} = \mathbf{K} \odot e^{i\langle \boldsymbol{\Theta}, \mathbf{P}'_t \rangle}. \qquad (10)$$

This "phase warping" strategy offers dual theoretical advantages. First, by modifying the model's "observation coordinate system" rather than the feature values themselves, it preserves the standard Independent and Identically Distributed (i.i.d.) Gaussian distribution of the pre-trained features, thereby fundamentally eliminating artifacts caused

by Out-of-Distribution shifts. Second, benefiting from the group operation properties of $\mathbf{M}_t$ on the Lie group, the phase evolution of positional encodings exhibits strict spectral continuity. Even when handling extreme camera movements like large-angle rotations or rapid zooms, the model maintains the coherence of high-frequency textures, effectively resolving common issues in traditional methods such as background tearing and structural breakage.

### 4.4. Inference-Time Geometric Guidance

To precisely estimate the coupled motion fields defined in Eq. (9), we perform test-time optimization. Instead of unconstrained tuning, we model the generation process as a collaborative evolution of the Lie algebra parameters $\boldsymbol{\xi} \in \mathfrak{sim}(2)$ and the correction field $\Delta\mathbf{P} \in \mathbb{R}^{T \times H \times W}$:

$$\hat{\boldsymbol{\xi}}, \Delta\hat{\mathbf{P}} = \arg\min_{\boldsymbol{\xi}, \Delta\mathbf{P}} \mathcal{L}_{\text{align}}\left(\mathbf{v}_\theta\left(t, \mathbf{z}_t \mid \mathbf{P}'_t\right), \mathbf{v}_{\text{ref}}(t)\right), \quad (11)$$

where $\mathbf{v}_\theta$ denotes the predicted denoising direction, and $\mathbf{v}_{\text{ref}} = (\mathbf{z}_t - \mathbf{z}_0^{\text{ref}})/\sigma_t$ is the reference denoising residual in the latent space. The alignment loss $\mathcal{L}_{\text{align}}$ adopts the hybrid objective similar to RoPECraft, combining spatial MSE for pixel fidelity and Fourier phase loss for structural integrity.

This formulation constructs a critical **Geometric Bottleneck**. The low-dimensional $\boldsymbol{\xi}$ (4 DoF) provides a significantly steeper gradient path, thereby rapidly locking the evolution trajectory of the global reference frame. Meanwhile, $\Delta\mathbf{P}$ relies on the coordinate transformation determined by $\boldsymbol{\xi}$ to be correctly mapped into the observation space. Consequently, the optimizer's convergence process first establishes the global transport operator, and subsequently, under the constraints of this dynamic coordinate system, refines the accompanying $\Delta\mathbf{P}$.

## 5. Experiments

### 5.1. Experimental Setup

#### 5.1.1. DATASETS & PROTOCOLS

We select the **DAVIS 2017** (Pont-Tuset et al., 2017) dataset as our testing benchmark, which contains rich and complex object motion and rigid camera motion. We utilize **Gemini** (Team et al., 2023) to generate two differentiated text prompts for each reference video. The final evaluation set comprises 200 "reference-generation" video pairs.

#### 5.1.2. BASELINES

To verify LieWarper's effectiveness, we compare it with five representative methods. **Go-with-the-Flow** (Burgert et al., 2025) guides motion by real-time warping of initial noise using reference flow. **RoPECraft** (Gokmen et al., 2025)

aligns trajectories by iteratively optimizing RoPE phase parameters. **DiTFlow** (Pondaven et al., 2025) directs the denoising process using reference flow features. **Motion-Clone** (Ling et al., 2024) clones motion trends by injecting reference temporal attention maps as priors. **SMM** (Yatim et al., 2024) leverages space-time correspondences in deep features for zero-shot transfer.

#### 5.1.3. EVALUATION METRICS

We evaluate performance across three dimensions. FTD (Gokmen et al., 2025) tracks long-term trajectory alignment, while Flow Score (via RAFT (Teed & Deng, 2020)) measures dense directional precision. Depth Correlation (Yang et al., 2024) and Warping Error (Lai et al., 2018) quantify 3D structural preservation and local stability via flow-based reconstruction. Additionally, CLIP Score and Temp-Consistency (Radford et al., 2021) evaluate text-video alignment and temporal coherence, ensuring that motion constraints do not compromise the quality of the generation.

#### 5.1.4. IMPLEMENTATION DETAILS

In our experiment, we use the open-sourced video generation model WAN-2.1-1.3B (Wan et al., 2025) as the base text-to-video generation model. All input reference videos are resized to a resolution of $480 \times 832$ with a fixed length of 49 frames. Optical flow extraction employs the pre-trained RAFT model. The HOSVD rank is set to 4. During the inference-time fine-tuning stage, we use the AdamW (Loshchilov & Hutter, 2017) optimizer to perform lightweight updates, and the number of iterations is set to 5 steps. All experiments are conducted on a single NVIDIA 40GB A100 GPU with 50 inference steps.

### 5.2. Comparison with State-of-the-Art

#### 5.2.1. QUANTITATIVE EVALUATION

Table 1 presents quantitative comparisons of LieWarper against baselines on the DAVIS dataset. The data indicates our method achieves breakthrough improvements in motion trajectory consistency and geometric rigidity while ensuring foundational generation quality.

**Motion Trajectory Fidelity.** This is the decisive metric for measuring whether the generated video successfully clones the motion patterns of the reference video. LieWarper achieves the best performance (0.2381) on the FTD metric, significantly lower than Go-with-the-Flow (0.2497). This indicates that compared to noise or feature injection methods, our global analytic solver strategy captures and transfers complex object trajectories more precisely. Furthermore, our Flow-Score reaches the highest across the board (0.6907). This demonstrates that the motion magnitude generated by LieWarper is the most robust, avoiding the

*Table 1.* **Quantitative Comparison on DAVIS2017 Dataset.** Bold denotes the best, and underline denotes the second best.

| Method | FTD ↓ | Flow-Score ↑ | Depth-Corr ↑ | Warp-Err ↓ | CLIP Score ↑ | Temp-Con ↑ |
|---|---|---|---|---|---|---|
| **Ours** | **0.2381** | **0.6907** | **0.8418** | 19.3053 | 0.3154 | 0.9753 |
| Go-with-the-Flow | 0.2497 | 0.6256 | 0.6758 | 21.1506 | **0.3242** | 0.9710 |
| RoPECraft | 0.2501 | 0.6771 | 0.8056 | 20.2687 | 0.3102 | 0.9627 |
| DiTFlow | 0.2576 | 0.3057 | 0.6319 | **14.5425** | 0.3152 | **0.9832** |
| MotionClone | 0.2529 | 0.6390 | 0.6454 | 49.8197 | 0.3075 | 0.9186 |
| SMM | 0.2548 | 0.6810 | 0.6515 | 31.0229 | 0.3137 | 0.9511 |

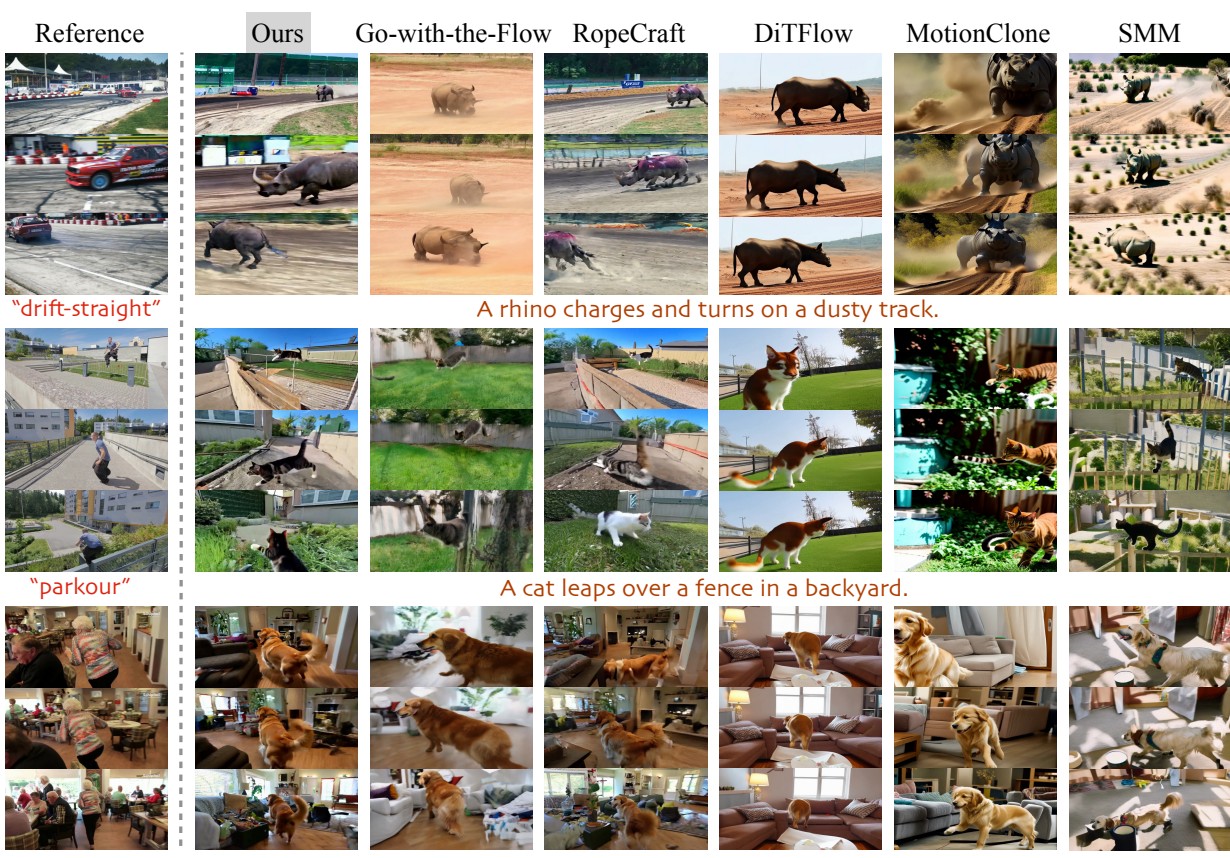

*Figure 4.* **Qualitative comparison of motion transfer. Best viewed with zoom in.**

"motion collapse" phenomenon seen in DiTFlow (0.3057), which circumvents errors by generating approximately static frames. We successfully preserve the full range of motion while replicating its trajectory with high precision.

**Geometric Rigidity & Structure Generation.** While ensuring consistent object trajectories, the rationality of camera ego-motion directly determines the physical realism of the generated video. LieWarper reaches 0.8418 on the Depth-Corr metric, surpassing all methods including RoPECraft. This proves that $\mathrm{Sim}(2)$ manifold constraints effectively guide the model to generate scene structures that conform to 3D perspective relationships. Regarding the Warp-Err metric, although DiTFlow has the lowest value,

its extremely low Flow-Score suggests it fails to generate effective motion. In contrast, LieWarper maintains an extremely low warping error (19.31) under the premise of high-dynamic generation, proving that we constrain the rigid transformation of the background during generation, avoiding non-physical liquefaction or shearing.

**Generation Quality.** For CLIP Score (0.3154) and Temp_Consistency (0.9753), LieWarper maintains competitive performance. This verifies that our geometric control module is compatible with the pre-trained diffusion model, applying strict motion constraints does not have a negative impact on the semantic alignment and temporal coherence.

*Table 2.* **Ablation Study.** We conduct a comprehensive analysis on (a) analytic modeling, (b) coupling mechanisms, and (c) optimization strategies. Default settings are marked in  gray .

| *(a)* **Analytic Modeling** | | |
|---|---|---|
| **Variant** | **Warp-Err ↓** | **Depth-Corr ↑** |
| w/o HOSVD | 20.1134 | 0.8103 |
| Affine (6-DoF) | 24.1749 | 0.7632 |
| Sim(2)(ours) | **19.3053** | **0.8418** |

| *(b)* **Warping Mechanism** | | |
|---|---|---|
| **Mechanism** | **FTD ↓** | **Flow-Score ↑** |
| Rigid Only | 0.3854 | 0.5531 |
| Non-Rigid Only | 0.2501 | 0.6771 |
| Manifold Modulation | **0.2381** | **0.6907** |

| *(c)* **Optimization Strategy** | | | |
|---|---|---|---|
| **Strategy** | **FTD ↓** | **Warp-Err ↓** | **Time (s)** |
| No Analytic | 0.4126 | 27.9324 | 146.37 |
| Analytic + 0 step | 0.3814 | 28.3225 | 154.28 |
| Analytic + 5 steps | 0.2381 | 19.3053 | 289.05 |
| Analytic + 10 steps | 0.2366 | 19.0134 | 533.25 |

### 5.2.2. QUALITATIVE EVALUATION

Figure 4 displays the results of challenging videos covering various compound motions. This qualitative assessment offers an intuitive perspective on trajectory and geometric consistency, complementing the statistical metrics.

In the *Drift-straight* scenario, baselines like RoPECraft lose coordinate tracking during rapid panning, causing the generated rhinoceros to drift out of the frame. Conversely, LieWarper enforces strict geometric constraints, faithfully mimicking the high-speed reference trajectory.

For the *Parkour* sequence, characterized by rapid compound movements involving running, jumping, and camera tracking, baselines fail to reconstruct the complete kinematic chain, often missing the vertical parabolic apex or the rhythm of the steps. LieWarper, leveraging its global skeleton constraint, captures these intricate dynamics, faithfully replicating both the rhythmic strides and the airborne trajectory of the reference motion.

In the reference video *Lady-running*, attention-based methods often cause the subject to result in disjointed background motion. With LieWarper, the camera appears physically locked to the generated subject. The background recession speed maintains perfect perspective correspondence with the dog, verifying our accurate replication of both camera ego-motion and object motion.

### 5.3. Ablation Study

To investigate the theoretical validity and component-wise effectiveness of LieWarper, we conduct a comprehensive ablation study. All experiments are conducted on the DAVIS dataset under a unified evaluation protocol.

### 5.3.1. EXPERIMENTAL SETTINGS

**Analytic Modeling** (Table 2a). To validate the necessity of Sim(2) manifold modeling, we compare with: **1) w/o HOSVD**, which uses raw noisy flow directly without HOSVD filtering; and **2) Affine (6-DoF)**, which relaxes the manifold constraint to general affine transformations (allowing shearing) to test rigid consistency.

**Warping Mechanism** (Table 2b). We evaluate our flow-guided warping mechanism against: **1) Rigid Only**, applying only the global rigid transformation $\mathbf{M}_t$ while ignoring local flow guidance; and **2) Non-Rigid Only**, removing the explicit rigid modeling and relying solely on the high-DoF flow field ($\mathbf{P}' = \mathbf{P} + \Delta\mathbf{P}$).

**Optimization Strategy** (Table 2c). We assess solver efficiency via: **1) No Analytic Init**, optimizing $\xi$ from scratch (random initialization) for 50 iterations; and **2) Analytic Init + $N$ steps**, utilizing our closed-form solution initialized with $N \in \{0, 5, 10\}$ fine-tuning steps.

### 5.3.2. ANALYSIS OF RESULTS

**Analytic Modeling (Table 2a).** Removing the HOSVD filter (*w/o Global Filter*) results in performance degradation across all metrics, verifying that spectral truncation is essential for suppressing noise in the solver input. Furthermore, the significant increase in Warping Error for the *Affine (6-DoF)* variant (19.31 → 24.17) confirms that allowing shearing distortions compromises background rigidity.

**Warping Mechanism (Table 2b).** *Rigid Only* yields a high FTD (0.3854), demonstrating that a purely rigid transformation fails to capture non-rigid foreground dynamics. Conversely, *Non-Rigid Only* relies solely on the dense residual; while it improves Flow-Scores, the lack of a global skeleton leads to trajectory drift. Our method achieves the lowest FTD, proving that optimal fidelity is achieved by fitting motion under rigid coordinate transformations.

**Optimization Strategy (Table 2c).** Random initialization (*No Analytic*) fails to converge effectively (Warp-Err 27.93). In contrast, our *Analytic* solution provides a robust starting point. Fine-tuning for just 5 steps dramatically reduces the error (28.32 → 19.31), while extending to 10 steps yields negligible gains, confirming 5 steps as the optimal balance.

## 6. Conclusion

We propose LieWarper, modeling video motion as coordinate evolution on the Sim(2) manifold rather than pixel displacement. By defining dynamics within this intrinsic geometry, the generated visual changes are no longer superficial distortions, but natural projections of a consistent physical evolution. This work offers a new perspective for enhancing the physical plausibility of generative video.

## Acknowledgements

This work was supported by the National Natural Science Foundation of China under Grant U23A20300.

## Impact Statement

This work improves geometric consistency in video motion transfer. While aiding creation, greater physical plausibility makes fakes harder to spot, worsening deepfake misuse and detection difficulty.

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

# A. Mathematical Derivations

In this section, we provide the rigorous mathematical derivations supporting the geometric framework presented in Section 4. We detail the construction of the geometric Jacobian, the closed-form solution for the Sim(2) exponential map including numerical stabilization, and a theoretical error analysis comparing our geodesic evolution against Euclidean linear interpolation.

## A.1. Derivation of the Geometric Jacobian

The geometric Jacobian $\mathbf{J}(\mathbf{u})$ establishes the linear mapping between the Lie algebra parameters $\boldsymbol{\xi} \in \mathfrak{sim}(2)$ and the instantaneous velocity field in the image plane. Consider a point in the 2D image plane with coordinates $\mathbf{x} = [x, y]^T$, represented in homogeneous coordinates as $\mathbf{u} = [x, y, 1]^T$. The Lie algebra element $\boldsymbol{\xi} = [\sigma, \omega, v_x, v_y]^T$ comprises the isotropic scaling rate $\sigma$, angular velocity $\omega$, and translational velocities $(v_x, v_y)$. The corresponding Lie algebra matrix representation $\boldsymbol{\xi}^\wedge \in \mathbb{R}^{3 \times 3}$ is constructed from the generators of the similarity group:

$$\boldsymbol{\xi}^\wedge = \begin{bmatrix} \sigma & -\omega & v_x \\ \omega & \sigma & v_y \\ 0 & 0 & 0 \end{bmatrix}. \tag{12}$$

By definition, the tangent vector (velocity) $\dot{\mathbf{u}}$ at point $\mathbf{u}$ induced by the infinitesimal transformation $\boldsymbol{\xi}$ is given by the matrix-vector product $\dot{\mathbf{u}} = \boldsymbol{\xi}^\wedge \mathbf{u}$. Expanding this product yields:

$$\dot{\mathbf{u}} = \begin{bmatrix} \sigma & -\omega & v_x \\ \omega & \sigma & v_y \\ 0 & 0 & 0 \end{bmatrix} \begin{bmatrix} x \\ y \\ 1 \end{bmatrix} = \begin{bmatrix} \sigma x - \omega y + v_x \\ \omega x + \sigma y + v_y \\ 0 \end{bmatrix}. \tag{13}$$

The optical flow velocity vector $\mathbf{v}_i = [\dot{x}, \dot{y}]^T$ corresponds to the first two components of $\dot{\mathbf{u}}$. To express $\mathbf{v}_i$ as a linear function of the parameter vector $\boldsymbol{\xi}$, we factorize the expression by isolating the elements of $\boldsymbol{\xi}$:

$$\mathbf{v}_i = \begin{bmatrix} \dot{x} \\ \dot{y} \end{bmatrix} = \underbrace{\begin{bmatrix} x & -y & 1 & 0 \\ y & x & 0 & 1 \end{bmatrix}}_{\mathbf{J}(\mathbf{u})} \begin{bmatrix} \sigma \\ \omega \\ v_x \\ v_y \end{bmatrix}. \tag{14}$$

Here, $\mathbf{J}(\mathbf{u}) \in \mathbb{R}^{2 \times 4}$ is the derived Geometric Jacobian, which varies spatially with the pixel coordinate $\mathbf{u}$, strictly satisfying the constraint in Eq. (5) of the main text.

## A.2. Closed-Form Exponential Map for Sim(2)

While the exponential map is formally defined by the power series $\exp(\mathbf{X}) = \sum_{k=0}^{\infty} \frac{\mathbf{X}^k}{k!}$, computing this series is computationally expensive and numerically unstable. We derive the analytic closed-form solution for Sim(2).

We decompose $\boldsymbol{\xi}^\wedge$ into a linear transformation block $\mathbf{A}$ and a translation block $\mathbf{b}$:

$$\boldsymbol{\xi}^\wedge = \begin{bmatrix} \mathbf{A} & \mathbf{b} \\ \mathbf{0} & 0 \end{bmatrix}, \quad \text{where} \quad \mathbf{A} = \begin{bmatrix} \sigma & -\omega \\ \omega & \sigma \end{bmatrix}, \quad \mathbf{b} = \begin{bmatrix} v_x \\ v_y \end{bmatrix}. \tag{15}$$

The matrix exponential $\mathbf{M} = \exp(\boldsymbol{\xi}^\wedge)$ then takes the form of an affine transformation matrix:

$$\mathbf{M} = \begin{bmatrix} e^{\mathbf{A}} & \mathbf{t} \\ \mathbf{0} & 1 \end{bmatrix}. \tag{16}$$

**Linear Transformation $e^{\mathbf{A}}$.** Observing that $\mathbf{A} = \sigma \mathbf{I} + \omega \mathbf{J}$ (where $\mathbf{J}$ is the $2 \times 2$ rotation generator), and since $\mathbf{I}$ and $\mathbf{J}$ commute, we have:

$$e^{\mathbf{A}} = e^{\sigma} \begin{bmatrix} \cos\omega & -\sin\omega \\ \sin\omega & \cos\omega \end{bmatrix}. \tag{17}$$

**Translation t.** The translation component is obtained by integrating the linear flow: $\mathbf{t} = \left( \int_0^1 e^{\mathbf{A}\tau} d\tau \right) \mathbf{b}$. For the general case where $\det(\mathbf{A}) \neq 0$ (i.e., $\sigma^2 + \omega^2 > 0$), the integral evaluates to:

$$\mathbf{t} = \mathbf{A}^{-1}(e^{\mathbf{A}} - \mathbf{I})\mathbf{b}. \tag{18}$$

Substituting the explicit inverse of $\mathbf{A}$, we obtain the analytical expression for $\mathbf{t}$.

**Numerical Stability near Identity.** A critical implementation detail arises when the motion approaches pure translation (i.e., $\sigma \to 0$ and $\omega \to 0$), causing $\mathbf{A}$ to become singular. Direct application of the closed-form inverse leads to numerical instability. To ensure differentiability and stability during optimization, we employ a second-order Taylor approximation when $\|\mathbf{A}\|_F < \epsilon$ (typically $\epsilon = 10^{-4}$):

$$\mathbf{t} \approx \left( \mathbf{I} + \frac{1}{2!}\mathbf{A} + \frac{1}{3!}\mathbf{A}^2 \right) \mathbf{b}. \tag{19}$$

This hybrid formulation guarantees that the exponential map is well-conditioned and fully differentiable across the entire parameter manifold.

### A.3. Error Analysis: Geodesic vs. Linear Interpolation

We theoretically quantify the geometric distortion introduced by Euclidean linear interpolation. Let $\mathbf{M}_0 = \mathbf{I}$ be the identity pose and $\mathbf{M}_1$ be a target pose involving a rotation by angle $\theta$. We compare the determinant of the transformation along the path, which serves as a proxy for volume conservation (rigid bodies must have unit scaling factor).

Geodesic Path (Ours): The motion evolves along the one-parameter subgroup. For a pure rotation, the scaling factor $s(t) \equiv 1$. Thus, $\det(\mathbf{M}_{geo}(t)) = 1$ for all $t \in [0, 1]$.

**Linear Path.** The interpolation is defined as $\mathbf{M}_{lin}(t) = (1 - t)\mathbf{I} + t\mathbf{M}_1$. Consider the case of a $\pi$ rotation (180 degrees), where $\mathbf{M}_1$'s upper-left block is $-\mathbf{I}$. At the midpoint $t = 0.5$:

$$\mathbf{R}_{lin}(0.5) = 0.5\mathbf{I} + 0.5(-\mathbf{I}) = \mathbf{0}. \tag{20}$$

At this point, $\det(\mathbf{R}_{lin}(0.5)) = 0$. Physically, this implies the entire image plane collapses to a single point before expanding back out, a phenomenon we term *Volume Collapse*. For a general rotation $\theta$, the scaling error is $E_{scale}(t) = 1 - \sqrt{(1 - t)^2 + t^2 + 2t(1 - t)\cos\theta}$. This error is non-zero for all $\theta \neq 0$, proving that linear interpolation inherently violates rigid body constraints.

## B. Implementation Details

### B.1. Analytic Solver via Spectral Tensor Contraction

Unlike traditional methods that rely on iterative robust estimators (e.g., RANSAC or IRLS) to handle optical flow noise, we leverage the spectral properties of rigid motion. Our solver consists of two stages: HOSVD Denoising and Analytic Contraction. The complete procedure is detailed in Algorithm 1.

---

**Algorithm 1** Analytic Rigid Motion Solver with HOSVD Filtering

---

1: **Input:** Raw Optical Flow Tensor $\mathcal{F} \in \mathbb{R}^{T \times H \times W \times 2}$, Geometric Jacobian Tensor $\mathcal{J}$
2: **Output:** Global Lie Algebra parameters $\boldsymbol{\xi}^*$
3: **Hyperparameter:** Temporal Rank Cutoff $k = 4$
4: {— Stage 1: Temporal Spectral Denoising —}
5: Unfold tensor along time: $\mathbf{M} \leftarrow \text{Reshape}(\mathcal{F}, [T, 2HW])$
6: Compute SVD: $\mathbf{U}, \boldsymbol{\Sigma}, \mathbf{V}^T \leftarrow \text{SVD}(\mathbf{M})$
7: Truncate Spectrum:
8: $\quad \hat{\mathbf{U}} \leftarrow \mathbf{U}[:, :k], \quad \hat{\boldsymbol{\Sigma}} \leftarrow \boldsymbol{\Sigma}[:k, :k], \quad \hat{\mathbf{V}} \leftarrow \mathbf{V}[:, :k]$
9: Reconstruct Low-Rank Flow:
10: $\quad \hat{\mathbf{M}} \leftarrow \hat{\mathbf{U}}\hat{\boldsymbol{\Sigma}}\hat{\mathbf{V}}^T$
11: $\hat{\mathcal{F}} \leftarrow \text{Reshape}(\hat{\mathbf{M}}, [T, H, W, 2])$
12: {— Stage 2: Tensor Contraction —}
13: Flatten spatial dims: $\hat{\mathbf{F}} \in \mathbb{R}^{N \times 2}, \mathbf{J} \in \mathbb{R}^{N \times 2 \times 4}$
14: $\mathcal{A} \leftarrow \sum_{i=1}^{N} \mathbf{J}_i^T \mathbf{J}_i \quad$ {Global Structure Tensor $\mathbb{R}^{4 \times 4}$}
15: $\mathbf{b} \leftarrow \sum_{i=1}^{N} \mathbf{J}_i^T \hat{\mathbf{F}}_i \quad$ {Projection Vector $\mathbb{R}^4$}
16: {— Stage 3: Closed-Form Solution —}
17: $\boldsymbol{\xi}^* \leftarrow \mathcal{A}^{-1} \mathbf{b} \quad$ {Linear Solve}
18: **return** $\boldsymbol{\xi}^*$

---

The temporal rank cutoff $k$ acts as a hyperparameter controlling the trade-off between trajectory fidelity and noise suppression. We set $k = 4$ based on the spectral analysis shown in Figure 5. Empirically, we observe that the first 4 temporal components account for the vast majority of the motion energy (representing the smooth global trajectory), while the residual components correspond to stochastic noise and temporal incoherence. Truncating at this point provides a temporally smoothed flow field for the subsequent analytic solver.

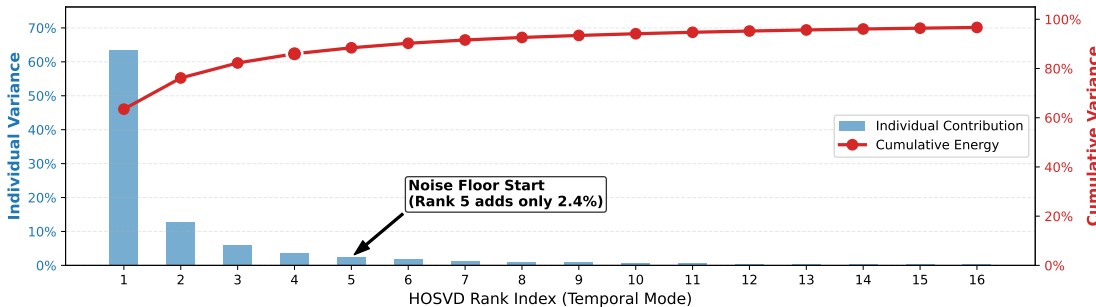

*Figure 5.* Scree Plot of Temporal Singular Values. The chart displays the individual (bar) and cumulative (line) explained variance ratios for the temporal modes of the motion tensor, averaged across all video sequences in the dataset.

### B.2. Detailed Optimization Protocols

In this section, we elaborate on the specific implementation protocols regarding the reference trajectory construction and the parameterization of the correction field.

**Reference Velocity Construction.** Consistent with Eq. (11) in the main text, the reference target $\mathbf{v}_{\text{ref}}$ is computed analytically based on the flow matching formulation. At each inference timestep $t$, we define the target vector field as the direction pointing from the current noisy latent $\mathbf{z}_t$ towards the clean reference latent $\mathbf{z}_0^{\text{ref}}$:

$$\mathbf{v}_{\text{ref}}(t) = \frac{\mathbf{z}_t - \mathbf{z}_0^{\text{ref}}}{\sigma_t}. \tag{21}$$

Here, $\sigma_t$ denotes the noise level. This term serves as the **ground-truth denoising direction** for the optimization. Physically, it signifies the instantaneous velocity required to transport the current noisy state $\mathbf{z}_t$ back to the reference video manifold. By

minimizing the discrepancy between the model prediction $\mathbf{v}_\theta$ and this analytic target $\mathbf{v}_{\text{ref}}$, we guide the generative trajectory to follow the reference motion dynamics without requiring auxiliary forward diffusion passes.

**Optimization Scope and Manifold Transport.** The geometric correction field $\Delta\mathbf{P}$ is parameterized as a cumulative learnable phase shift sequence, optimized progressively during the early inference steps. To prevent overfitting to high-frequency reference texture and to reduce memory overhead, $\Delta\mathbf{P}$ is **shared across all transformer layers and attention heads**. Crucially, we do not optimize $\Delta\mathbf{P}$ as a static Eulerian map. Instead, at each step, the learned parameter is subjected to the **Lie group manifold transport** before modulation:

$$\Delta\mathbf{P}'(t) = \mathcal{W}\left(\Delta\mathbf{P}^{(t)}; \mathbf{M}_t\right). \tag{22}$$

Mathematically, the operator $\mathcal{W}(\mathbf{X}; \mathbf{M})$ implements a differentiable backward warping mechanism. Given a feature map $\mathbf{X}$ and a rigid transformation $\mathbf{M} \in \text{Sim}(2)$, the warped value at pixel coordinate $\mathbf{u}$ is obtained via bilinear sampling:

$$\mathcal{W}(\mathbf{X}; \mathbf{M})[\mathbf{u}] = \text{BilinearSample}(\mathbf{X}, \mathbf{M}^{-1}\mathbf{u}), \tag{23}$$

where $\mathbf{M}^{-1}\mathbf{u}$ represents the coordinate projection from the current frame back to the canonical reference frame.

### B.3. Evaluation Metric Details

The detailed calculation protocols are as follows:

**Flow Score (Directional Fidelity).** To evaluate the dense directional alignment between the generated video and the reference video, we utilize the **RAFT** (Teed & Deng, 2020) model. For each corresponding frame pair, we extract the dense optical flow maps $\mathbf{F}_{\text{ref}}, \mathbf{F}_{\text{gen}} \in \mathbb{R}^{H \times W \times 2}$. The score is defined as the pixel-wise Cosine Similarity between the flow vectors:

$$\text{Score}_{\text{flow}} = \frac{1}{|\mathcal{M}|} \sum_{p \in \mathcal{M}} \frac{\mathbf{F}_{\text{ref}}(p) \cdot \mathbf{F}_{\text{gen}}(p)}{\|\mathbf{F}_{\text{ref}}(p)\|\|\mathbf{F}_{\text{gen}}(p)\| + \epsilon}, \tag{24}$$

where $\mathcal{M}$ is a dynamic validity mask. To focus on meaningful motion, we exclude static regions by filtering out pixels where the flow magnitude is below a threshold ($\tau = 0.1$) in either the reference or generated frames. The final metric is averaged across all frame pairs.

**Depth Correlation (Structural Preservation).** We assess the 3D structural integrity of the generated content using the Depth Anything (Yang et al., 2024) model ('depth-anything-small'). For each frame $t$, we estimate the monocular depth maps $D_{\text{ref}}^{(t)}$ and $D_{\text{gen}}^{(t)}$. The generated depth map is resized to match the reference resolution. We then flatten both maps into 1D vectors and compute the Pearson Correlation Coefficient (PCC):

$$\text{Corr}_{\text{depth}} = \frac{\text{Cov}(D_{\text{ref}}, D_{\text{gen}})}{\sigma_{D_{\text{ref}}} \sigma_{D_{\text{gen}}}}. \tag{25}$$

A higher correlation indicates that the generated video successfully preserves the relative depth relations and 3D perspective of the reference scene.

**Warping Error (Local Temporal Stability).** To measure the local temporal coherence and stability of the generated video itself (independent of the reference), we calculate the Warping Error using traditional optical flow. Specifically, we use the Farneback algorithm (via OpenCV) to compute the flow $\mathbf{F}_{t-1 \to t}$ between consecutive generated frames $I_{t-1}$ and $I_t$. We then warp frame $I_{t-1}$ using this flow to obtain a prediction $\hat{I}_t$. The Warping Error is defined as the Mean Absolute Error (MAE) between the warped frame and the actual frame:

$$E_{\text{warp}} = \frac{1}{T-1} \sum_{t=1}^{T-1} \|I_t - \mathcal{W}(I_{t-1}, \mathbf{F}_{t-1 \to t})\|_1. \tag{26}$$

Lower values indicate smoother local dynamics and fewer temporal artifacts such as flickering or texture swimming.

## C. Limitations

While LieWarper achieves robust motion transfer, we identify two primary limitations, regarding which we conducted specific attempts and detailed discussions.

**Trade-off between Temporal Resolution and Solver Stability.** We model video motion as a global geodesic evolution governed by a constant Lie algebra $\boldsymbol{\xi}^*$. As shown in Figure 6 (second row), although the coordinate evolution term $\Delta P$ attempts to track rapid alternating maneuvers (e.g., sharp left-right switching), the strict regularity imposed by the global constant-velocity assumption causes the view frustum to "lag" behind the erratic trajectory. Consequently, elements that should temporarily exit the field of view remain incorrectly visible. A natural critique is: why not adopt a time-variant parameterization $\boldsymbol{\xi}_t$ (e.g., a piecewise linear model) to capture multi-segment motion?

To avoid trajectory discontinuities caused by simple temporal slicing, we explored a **sliding window with overlap** strategy. Specifically, to ensure alignment with the VAE compression ratio, we divided the video sequence into overlapping windows of size $w = 16$ and stride $s = 4$, aiming to solve for the local Lie algebra $\boldsymbol{\xi}_t$ at time $t$. This process is modeled as minimizing the local energy function:

$$E_t(\boldsymbol{\xi}) = \sum_{k=t}^{t+w-1} \sum_{i \in \Omega} \|J_{k,i}\boldsymbol{\xi} - \hat{\mathcal{F}}_{k,i}\|_2^2 \tag{27}$$

The analytic solution is given by $\boldsymbol{\xi}_t^* = \mathcal{A}_t^{-1}\mathbf{b}_t$, where the local structure tensor is defined as $\mathcal{A}_t = \sum_{k=t}^{t+w-1} \sum_i J_{k,i}^T J_{k,i}$.

However, experimental results indicate that such time-variant parameterization introduces severe **temporal discontinuity**. Independent optimization within local windows leads to inconsistent parameter evolution during camera movement. This lack of global context manifests as a critical semantic failure, as shown in Figure 6 (third row): although the local solver allows objects to correctly exit the frame, the drastic variations in parameters cause discontinuities in the optimization process, preventing the diffusion model from maintaining identity persistence.

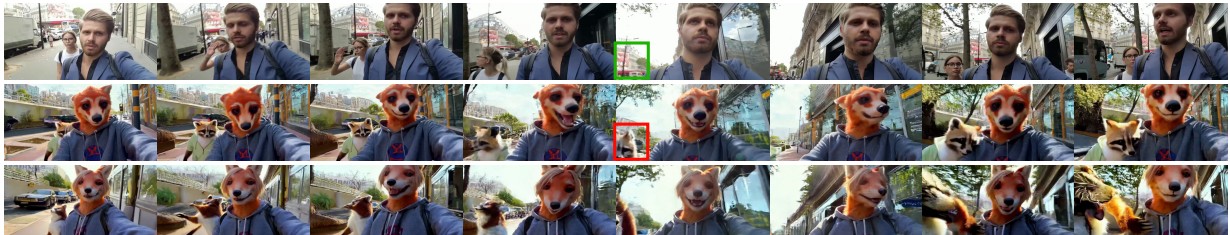

A fox and a raccoon were walking on the street, holding cameras and taking selfies.

*Figure 6.* **Trade-off between temporal resolution and solver stability. Global Solver (Second Row):** While maximizing geometric rigidity, the constant-velocity assumption causes a "lag" effect, where elements that should exit the field of view remain incorrectly visible (highlighted by the red box). **Local Sliding Window (Third Row):** While allowing objects to exit, the independent window optimization causes **parameter discontinuity**. This disrupts the diffusion process, resulting in severe **identity inconsistency** when the object re-enters the frame.

**Planar Approximation Sim(2) vs. SE(3).** Admittedly, while the Sim(2) manifold effectively characterizes the global geometric rigidity of camera projection, it inherently models motion as a planar homography transformation, thus lacking the capacity to handle complex $Z$-dependent parallax. As shown in Figure 7 (second row), this limitation leads to structural distortions in scenarios with extreme depth variations (e.g., roller coasters), where the planar approximation fails to decouple foreground and background motion.

To investigate the efficacy of explicit 3D control, we explored lifting the manifold to the full 6-DoF SE(3) group. Lacking ground truth camera intrinsics $\mathbf{K}_{gt}$, we assume a fixed field-of-view $\theta_{fov}$ (set to $60°$ in experiments) to construct an approximate intrinsic matrix $\mathbf{K}_{est}$. Assuming a latent feature map resolution of $H \times W$, the normalized focal length is defined as $f_{est} = W/(2\tan(\theta_{fov}/2))$. We utilize the relative depth map $Z(\mathbf{u})$ estimated by Depth Anything V2 (Yang et al., 2024) to perform back-projection, obtaining pseudo-world coordinates $\mathbf{P}_{pseudo} = Z(\mathbf{u}) \cdot \mathbf{K}_{est}^{-1}\mathbf{u}$. Subsequently, we apply a rigid transformation $\mathbf{T} = \exp(\boldsymbol{\xi}^\wedge)$ in the SE(3) Lie group, and re-project the transformed 3D points $\mathbf{P}'_{pseudo}$ back to the 2D image plane to obtain new coordinates $\mathbf{u}'$. The resulting SE(3) geometric residual field $\Delta\mathbf{P}_{se3}(\mathbf{u}) = \mathbf{u}' - \mathbf{u}$ is used to replace the Sim(2) evolution term in the main method. Specifically, we define the updated coordinate field as:

$$P'_{se3} = P + \Delta P_{se3} \tag{28}$$

This field is subsequently used to modulate the Query and Key vectors via Eq 10.

However, empirical results indicate that this scheme leads to severe generation collapse (as shown in Figure 7, third row). We attribute this failure to the **ill-conditioned nature** of the re-projection function under noisy depth estimation. Consider the depth estimate containing additive noise $Z = Z_{gt} + \delta Z$. Under a simplified assumption of lateral camera translation $t_x$, the re-projected horizontal coordinate $u'$ can be approximated as $u' \approx (u - c_x) + \frac{f_{est}t_x}{Z}$. This relationship indicates that the translation term is inversely proportional to depth, and its partial derivative with respect to $Z$ is:

$$\frac{\partial u'}{\partial Z} = -\frac{f_{est}t_x}{Z^2} \tag{29}$$

This result reveals that the coordinate error is proportional to the square of the inverse depth: even imperceptible depth noise $\delta Z$ in the near-field (small $Z$) is amplified into catastrophic jumps in the coordinate field. Since RoPE is extremely sensitive to the **Spectral Continuity** of the phase $\theta'$, such high-frequency phase discontinuities induced by depth noise directly destroy the local correlation in Self-Attention, resulting in texture tearing and the collapse of generation quality.

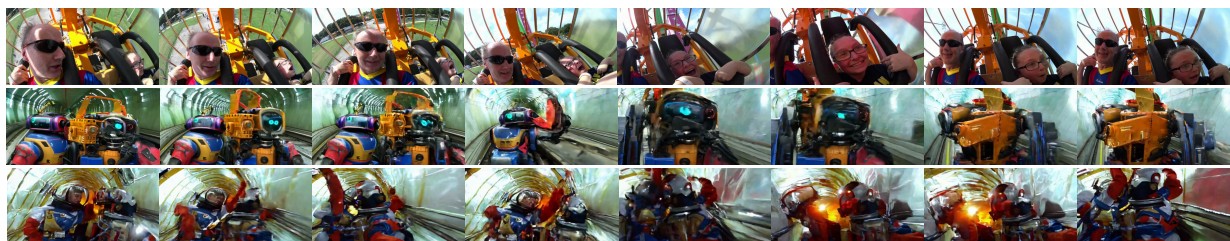

Two robots are shuttling through a time tunnel on a train.

*Figure 7.* **Ablation on Manifold Choice. Planar Sim(2) Limitation (Second Row):** The planar homography assumption fails to model complex parallax in scenes with extreme depth variations. This inability to decouple foreground and background motion results in noticeable structural distortion. **Explicit SE(3) Collapse (Third Row):** While SE(3) theoretically handles parallax, the implementation via re-projection leads to severe texture tearing. Theoretically, depth noise is amplified by the inverse square of depth ($1/Z^2$), creating high-frequency discontinuities that break the spectral coherence of RoPE.

**Future Directions.** To mitigate temporal discontinuity, future work should transition from discrete window solving to global trajectory modeling (e.g., B-Splines). Instead of patching independent steps, parameterizing motion as a continuous curve ensures smoothness by design. However, this requires abandoning our efficient linear analytic solver in favor of a computationally intensive iterative optimizer.

Regarding SE(3) control, we face two fundamental hurdles. First is scale ambiguity: reliance on relative depth leads to unnatural "sliding" during camera translation, requiring metric depth to fix the absolute scale. Second is manifold mismatch: the diffusion model operates in a compressed latent space, not a physical 3D space. Rigid Euclidean transformations do not map linearly to this manifold, necessitating learnable geometric adapters instead of direct mathematical warping.

## D. Robustness Analysis of Optical Flow Estimation

In the main paper, we utilize RAFT as the default optical flow estimator. To investigate whether LieWarper's performance stems from the robust geometric modeling or merely the quality of the upstream flow estimator, we conduct a comprehensive robustness analysis. This section evaluates the system's sensitivity to (1) synthetic noise injection and (2) varying optical flow backbones.

### D.1. Sensitivity to Noise Injection

To simulate challenging conditions such as motion blur or low-light sensor noise, we introduce Additive White Gaussian Noise (AWGN) to the raw optical flow field $\mathcal{F}$ before it enters our pipeline. The noisy flow is defined as $\mathcal{F}_{noisy} = \mathcal{F} + \mathcal{N}(0, \sigma^2)$, where $\sigma$ represents the noise intensity in pixels. We compare LieWarper against the strongest baseline, Go-with-the-Flow, which directly warps noise using pixel-wise flow.

As shown in Table 3, baseline methods exhibit a sharp performance drop as noise increases, confirming their susceptibility to pixel-level errors. In contrast, LieWarper demonstrates significant resilience. Even at high noise levels ($\sigma = 5.0$), our

*Table 3.* Performance degradation under varying noise levels ($\sigma$). We compare our method against the strongest baseline, Go-with-the-Flow, under Additive White Gaussian Noise (AWGN) injected into the optical flow field.

| Noise Level ($\sigma$) | Method | FTD ↓ | Warp-Err ↓ | Depth-Corr ↑ | Relative Perf. Drop |
|---|---|---|---|---|---|
| $\sigma = 0$ (Clean) | Go-with-the-Flow | 0.2497 | 21.15 | 0.6758 | - |
| | **LieWarper (Ours)** | **0.2381** | **19.30** | **0.8418** | - |
| $\sigma = 2.0$ px | Go-with-the-Flow | 0.2654 | 22.48 | 0.6332 | -6.3% |
| | **LieWarper (Ours)** | **0.2464** | **19.98** | **0.8123** | **-3.5%** |
| $\sigma = 5.0$ px | Go-with-the-Flow | 0.2789 | 23.62 | 0.5967 | -11.7% |
| | **LieWarper (Ours)** | **0.2579** | **20.90** | **0.7719** | **-8.3%** |

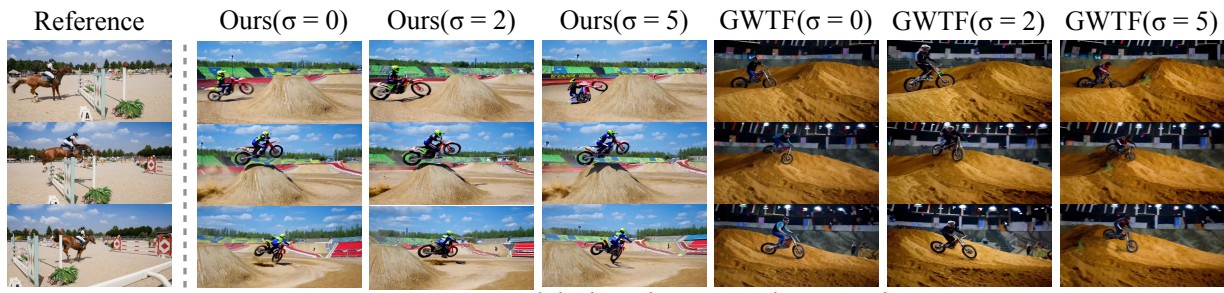

Reference    Ours($\sigma = 0$)    Ours($\sigma = 2$)    Ours($\sigma = 5$)    GWTF($\sigma = 0$)    GWTF($\sigma = 2$)    GWTF($\sigma = 5$)

"horsejump-high"      A motocross bike launches over a dirt mound in an arena.

*Figure 8.* Visualization under different optical flow noises.

method maintains a high Depth-Corr.

## D.2. Impact of Different Flow Estimators

To decouple our contribution from the specific choice of RAFT, we evaluate LieWarper using a traditional dense optical flow method (Farneback) and a different deep learning-based estimator (FlowFormer).

*Table 4.* Evaluation with different flow backbones. We test LieWarper using a traditional method (Farneback), the default estimator (RAFT), and a SOTA transformer (FlowFormer) to verify that our performance is not solely dependent on the flow estimator's quality.

| Flow Backbone | FTD ↓ | Flow-Score ↑ | Depth-Corr ↑ | Observations |
|---|---|---|---|---|
| Farneback (Traditional) | 0.2650 | 0.5820 | 0.7840 | Slightly jittery, but global perspective remains locked. |
| RAFT (Default) | 0.2381 | 0.6907 | 0.8418 | Balanced performance across all metrics. |
| FlowFormer (SOTA) | **0.2365** | **0.6950** | **0.8450** | Marginal improvement over RAFT; confirms convergence. |

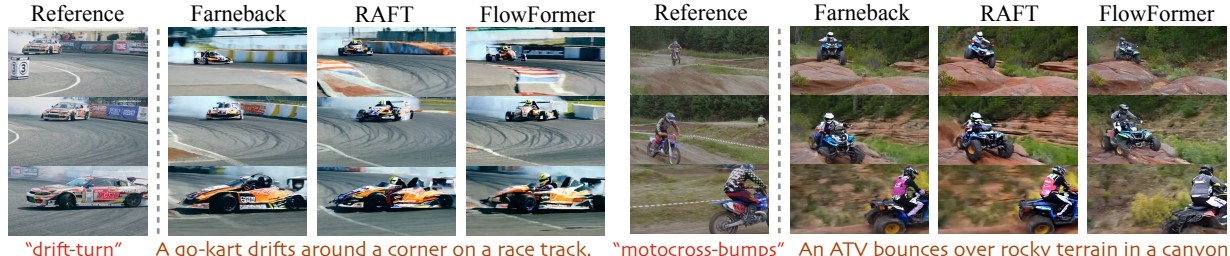

Reference    Farneback    RAFT    FlowFormer      Reference    Farneback    RAFT    FlowFormer

"drift-turn"    A go-kart drifts around a corner on a race track.    "motocross-bumps"    An ATV bounces over rocky terrain in a canyon.

*Figure 9.* Visualization of our method under different optical flow models.

Remarkably, even when using the non-deep Farneback algorithm, LieWarper successfully recovers a coherent global camera trajectory (Depth-Corr 0.7840). This proves that our Analytic $Sim(2)$ Modeling is the primary driver of geometric consistency, rather than the neural network prior of the flow estimator. The system extracts the "dominant rigid motion" even from noisy, imperfect flow fields.

# E. User Study

While quantitative metrics provide objective measures, they do not fully align with human perception, especially regarding the naturalness of motion and structural consistency. To comprehensively evaluate the visual quality of LieWarper, we conducted a rigorous **Two-Alternative Forced Choice (2AFC)** user study.

**Protocol.** We invited 30 independent evaluators. We randomly selected 25 video pairs from the test set. For each pair, the evaluator was presented with the reference video, the result generated by LieWarper, and the result from a baseline method (displayed in random order to prevent bias). Evaluators were asked to select the preferred video based on two criteria:

- **Motion Fidelity:** Which video better preserves the trajectory and motion dynamics of the reference?

- **Visual Quality:** Which video exhibits fewer artifacts (e.g., distortion, flickering, structural collapse)?

**Results.** We collected a total of 750 valid pairwise judgments. Table 5 summarizes the results. Our method consistently outperforms all baselines by a significant margin. In particular, LieWarper achieves an overall preference rate of 79.8% in motion fidelity and 78.6% in visual quality, calculated over all valid pairwise judgments rather than by averaging across baselines. More visualization results are shown in Figure 11.

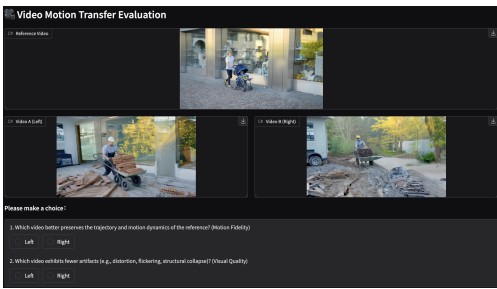

*Figure 10.* **User Study Interface.** The web-based 2AFC evaluation platform used in our study.

*Table 5.* **User Preference Rates.** Values indicate the percentage of votes favoring *LieWarper* over the baseline.

| Baseline | Motion | Quality |
|---|---|---|
| vs. Go-with-the-Flow | 81.8% | 86.2% |
| vs. RoPECraft | 73.2% | 72.4% |
| vs. DiTFlow | 83.5% | 82.4% |
| vs. MotionClone | 83.2% | 76.4% |
| vs. SMM | 82.2% | 82.6% |
| **Overall** | **79.8%** | **78.6%** |

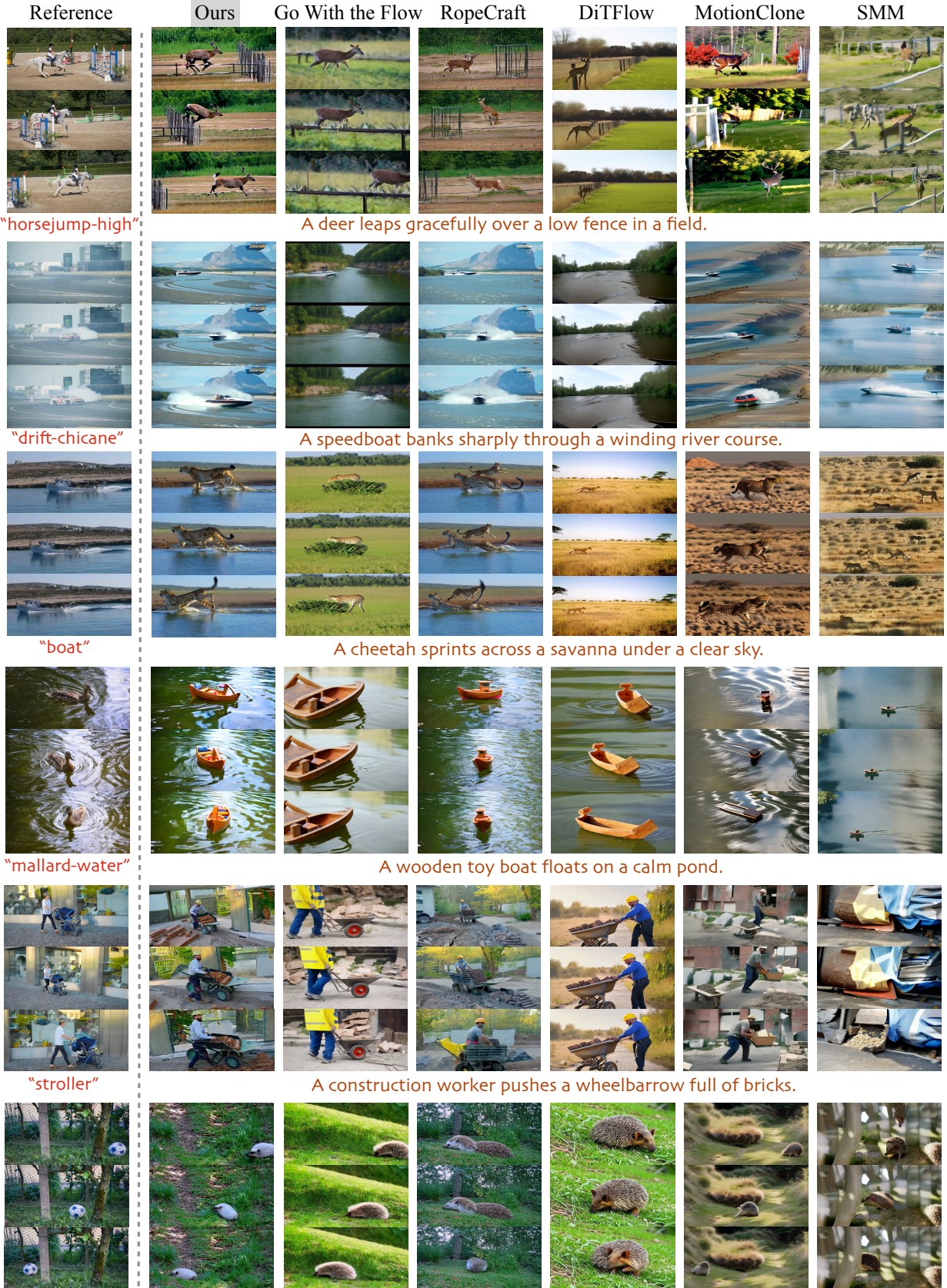

*Figure 11.* **Additional qualitative comparison of motion transfer. Best viewed with zoom in.**

