# OpenReview forum: "LieWarper: Geometry-Aware Motion Transfer via Lie Algebra"
_ICML.cc/2026/Conference — ICML 2026 regular_

### Official Review · Reviewer_WKP5 · 2026-03-09

**Soundness:** 3
**Presentation:** 3
**Significance:** 3
**Originality:** 2
**Overall Recommendation:** 4
**Confidence:** 2

**Summary:**

This paper proposes LieWarper, a training-free framework for video motion transfer that models motion as coordinate evolution on the Sim(2) Lie group manifold instead of pixel displacement. The method introduces an analytic solver based on HOSVD-filtered optical flow to estimate global similarity transformations, and combines it with flow-guided phase modulation to capture local non-rigid motion. An inference-time optimization with a geometric bottleneck further stabilizes motion transfer. Experiments on DAVIS 2017 show improvements over several training-free baselines in trajectory fidelity and geometric consistency.

**Compliance With Llm Reviewing Policy:**

Affirmed.

**Final Justification:**

The author's rebuttal addressed my concerns, I would keep my rating.

**Key Questions For Authors:**

1. For the SE(3) failure: Did you attempt depth map preprocessing (median filtering, edge-aware smoothing) to mitigate the 1/Z² noise sensitivity before re-projection?

2. For optimization efficiency: Table 2c shows 10 steps nearly doubles time with marginal gain—does this indicate a rugged loss landscape? Was L-BFGS or second-order methods explored?

3. For semantic alignment: Does strict geometric constraint suppress creative motion (e.g., "melting clock walking")? How do you interpret the CLIP Score drop?

**Limitations:**

Yes

**Strengths And Weaknesses:**

## Strengths:
1. Rigorous geometric foundation: The Sim(2) manifold formulation with closed-form exponential map and geometric Jacobian provides theoretical guarantees against volume collapse and shearing artifacts. Mathematical derivations in Appendix A are thorough.

2. Practical training-free design: No model fine-tuning required; control achieved through RoPE manipulation and lightweight inference optimization (~289s/video, 5 steps), making it accessible for practitioners.

3. Comprehensive validation: Ablations systematically verify components—HOSVD filtering vs. raw flow, Sim(2) vs. affine (6-DoF), rigid/non-rigid/hybrid warping, and analytic initialization vs. random optimization.
Noise robustness: Appendix D shows graceful degradation under σ=5px flow noise (vs. sharp drops in baselines), proving geometric constraints act as effective regularization.

## Weaknesses:
1. Fundamental 3D limitation: The planar Sim(2) assumption fails on extreme parallax (e.g., roller coasters). The SE(3) extension attempt failed due to depth noise amplification—this is not a minor implementation issue but a capability boundary of the current framework.

2. Unresolved temporal trade-off: Global constant-velocity causes "lag" on erratic motions; sliding windows enable local flexibility but destroy identity consistency (Figure 6). This intrinsic tension between geometric rigidity and temporal adaptability is acknowledged but not resolved.

3. Baseline asymmetry: Comparison limited to training-free methods; no evaluation against fine-tuned controllers (MotionCtrl, CameraCtrl) despite comparable inference cost. The Pareto frontier of quality vs. cost is unclear.

---

> ### Author Rebuttal · Authors · 2026-03-31
>
> We appreciate your recognition of the rigor of the manifold geometric foundations, mathematical derivations, and practicality of this research. In response to your questions, we provide the following clarifications.
>
> D1: Baseline Asymmetry
>
> MotionCtrl [1] and CameraCtrl [2] require explicit camera trajectory parameters and focus more on camera control.
>
> (1) We have included additional comparisons with MotionCtrl and CameraCtrl on the RealEstate10k dataset. Quantitative results and visualizations are available on the anonymous project page: https://anonymous.4open.science/r/Anonymous-repository-of-Liewarper.
>
> (2) We outperform the fine-tuning-based MotionCtrl and CameraCtrl in camera trajectory consistency (CamMC), even though they require explicit trajectory inputs. We achieved optimal results in directional accuracy (Flow Score) and 3D structure preservation (Depth Correlation), while securing sub-optimal performance in local stability (Warping Error) and generation quality.
>
> D2: 3D Limitations
>
> (1) As discussed in Appendix C, the lack of intrinsic parameters causes $SE(3)$ re-projection errors to be amplified. We attempted median filtering, but the generation results still collapsed. The fundamental reason is that filtering cannot eliminate the scale ambiguity inherent in monocular depth estimation. Minute errors are distorted by the amplification of the $1/Z^2$ partial derivative term, which disrupts the phase spectral continuity of the Rotary Positional Embeddings (RoPE).
>
> (2) $Sim(2)$ includes a scale parameter, which we utilize to implicitly model visual depth zooming (push/pull), supplemented by residual fields to absorb local parallax. Experiments on the RealEstate10k dataset demonstrate that in scenes with extensive camera movement and depth variations, our method exhibits superior motion fidelity and generation quality.
>
> D3: Temporal Adaptability
>
> We imposed smoothing regularization at the junctions of adjacent sliding windows by penalizing the first-order difference between the local Lie algebras of adjacent windows ($\|\xi_{t+s} - \xi_t\|_2^2$), thereby eliminating inconsistencies caused by independent solving. The visualization page showcases the results after incorporating this constraint. In the ‘video2:walking’ case from the DAVIS dataset, the identity drift previously triggered by window switching was eliminated. In the ‘video3:bmx-bumps’ case (which involves the subject exiting the frame, re-entering, and severe physical occlusion), the imposition of this continuity constraint effectively suppressed identity fluctuations and ensured robust structural coherence across window boundaries.
>
> D4: Optimization Efficiency
>
> We compared different optimization strategies across the first 10 steps to intuitively demonstrate efficiency.
>
> | Optimization Step | 1 | 2 | 3 | 4 | 5 | 6 | 7 | 8 | 9 | 10 |
> | :--- | :--- | :--- | :--- | :--- | :--- | :--- | :--- | :--- | :--- | :--- |
> | AdamW, w/o Init | 0.5332 | 0.5120 | 0.4784 | 0.4342 | 0.4151 | 0.4023 | 0.3989 | 0.3990 | 0.3952 | 0.3941 |
> | **AdamW, Ours** | 0.4316 | 0.3952 | 0.3810 | 0.3785 | 0.3772 | 0.3768 | 0.3765 | 0.3761 | 0.3759 | 0.3755 |
> | L-BFGS | 0.5332 | 0.5218 | 0.4860 | 0.4540 | 0.4257 | 0.4355 | 0.4492 | 0.4687 | 0.4863 | 0.4941 |
>
> (1) A lack of priors leads to slow convergence. Our initial loss (0.4316) is already superior to the baseline at its 4th step. The model enters a convergence plateau in only 4 steps; thus, the diminishing marginal returns are not due to a "rugged" optimization landscape.
>
> (2) We experimented with the L-BFGS optimization method. However, its line search mechanism requires multiple evaluations of the 1.3B Transformer within a single iteration, causing inference time to surge (996.5 seconds). Furthermore, its convergence performance is poor; it is prone to stagnating in local sub-optima or experiencing numerical instability due to the significant scale disparity between Lie parameters and local phases.
>
> D5: Semantic Alignment
>
> (1) Geometric constraints impose a slight regularization on the diffusion model's space for semantic divergence. However, because our manifold constraints are applied from a global perspective, creative motion is still derived from the data distribution of the pre-trained model itself.
>
> (2) Methods lacking geometric constraints allow the diffusion model to freely evolve toward "CLIP-friendly" results during generation. Visual results fully demonstrate our generated videos' high fidelity to the text prompts, indicating that the slight dip in CLIP scores is not a regression in generation quality, but rather a reasonable alignment cost for defending against geometric distortion.
>
> [1] Wang, Zhouxia, et al. "Motionctrl: A unified and flexible motion controller for video generation." ACM SIGGRAPH 2024.
>
> [2] He H, Xu Y, Guo Y, et al. "Cameractrl: Enabling camera control for video diffusion models" ICLR 2025.

---

> > ### Author Rebuttal · Reviewer_WKP5 · 2026-04-03
> >
> > My concerns have been adequately addressed.

---

> > > ### Author Response · Authors · 2026-04-06
> > >
> > > We are honored to have received your recognition and sincerely appreciate the time and effort you have invested.

---

### Official Review · Reviewer_d4cj · 2026-03-12

**Soundness:** 2
**Presentation:** 3
**Significance:** 3
**Originality:** 3
**Overall Recommendation:** 4
**Confidence:** 3

**Summary:**

This paper studies video motion transfer, aiming to generate new videos whose content follows the motion trajectories of a reference video. The authors observe that existing methods often model motion as simple pixel displacement in Euclidean space, which can cause geometric artifacts such as shearing and perspective collapse under complex camera and object motions. To address this limitation, they propose LieWarper, a geometry-aware motion transfer framework that models motion as coordinate evolution on the Sim(2) Lie group manifold. The method analytically estimates global camera motion from optical flow to construct a rigid reference frame, and then introduces a flow-guided phase warping mechanism to integrate local non-rigid dynamics while preserving global geometric consistency.

**Compliance With Llm Reviewing Policy:**

Affirmed.

**Final Justification:**

The authors have addressed the concerns raised in my review during the rebuttal phase, and I have increased my score accordingly.

**Key Questions For Authors:**

1. To further validate the generalizability of the proposed framework, experiments on additional benchmarks with diverse motion patterns and camera movements would provide stronger evidence that the Sim(2)-based formulation generalizes beyond the DAVIS2017 dataset.

2. Providing supplementary videos or animated visualizations would better demonstrate the temporal stability and motion consistency of the generated sequences, allowing readers to more clearly observe the improvements over baseline methods.

3. Providing a more detailed analysis of the experimental results, particularly for metrics such as CLIP Score and Temp-Consistency, would help clarify the behavior of the method and better explain the observed performance differences compared to other approaches.

**Limitations:**

The method relies on accurate optical flow estimation and assumes global motion can be modeled by Sim(2) transformations, which may limit its robustness in scenarios with complex 3D camera motion or inaccurate flow estimation. Further experimental analysis on such cases would help better evaluate the robustness of the proposed approach.

**Strengths And Weaknesses:**

### Summary of Strengths

1. **Clear geometric formulation for motion transfer.**
   The paper introduces a geometry-aware framework that models motion evolution on the Sim(2) Lie group manifold rather than treating motion as simple pixel displacement. This formulation provides a principled way to decouple global camera motion from local non-rigid dynamics.

2. **Analytic motion estimation and stable optimization.**
   The proposed analytic solver with HOSVD filtering enables robust estimation of global motion parameters from noisy optical flow.

3. **Clear and well-structured presentation.**

### Summary of Weaknesses

1. Experiments are mainly conducted on the DAVIS2017 dataset. Additional evaluations on more diverse video datasets or real-world scenarios with different motion patterns would help better validate the generalizability of the proposed approach.

2. While the visual comparisons appear convincing, providing supplementary video results would better demonstrate the temporal consistency and stability of the generated motion over time.

3.  Although LieWarper performs well on motion-related metrics, its CLIP Score and Temp-Consistency do not achieve the best performance among the compared methods. A more detailed discussion analyzing these results would help readers better understand the trade-offs between geometric motion control and generation quality.

---

> ### Author Rebuttal · Authors · 2026-03-31
>
> We appreciate your positive assessment of our manifold geometric modeling, the design of the analytical solver, and the logical consistency of our overall framework. In response to your questions, we provide a comprehensive reply below.
>
> C1: Dataset Generalization
>
> (1) We have included two additional comparison methods and conducted experiments on the RealEstate10k dataset. The quantitative metrics are as follows:
>
> ###
>
> | Method           | CamMC↓        | Flow-Score↑  | Depth-Corr↑  | Warp-Err↓     | CLIP Score↑  | Temp-Con↑    |
> | ---------------- | ------------- | ------------ | ------------ | ------------- | ------------ | ------------ |
> | **Ours**         | **0.0282**    | **0.543**    | **0.537**    | *11.964* | *0.302* | *0.978* |
> | Go-with-the-Flow | 0.0304        | *0.461* | 0.465        | 17.119        | 0.284        | 0.974        |
> | RopeCraft        | 0.0298        | 0.424        | 0.505        | 13.024        | 0.284        | 0.977        |
> | ditflow          | 0.0648        | 0.428        | 0.361        | **10.604**    | **0.310**    | **0.979**    |
> | MotionClone      | 0.0369        | 0.351        | 0.387        | 12.651        | 0.254        | 0.932        |
> | SMM              | 0.0353        | 0.452        | *0.509* | 24.667        | 0.279        | 0.966        |
> | MotionCtrl       | *0.0302* | 0.437        | 0.374        | 19.988        | 0.298        | 0.960        |
> | CameraCtrl       | 0.0313        | 0.384        | 0.415        | 17.497        | 0.294        | 0.962        |
>
> (2) The results indicate that our method outperforms MotionCtrl and CameraCtrl—which require explicit input of trajectory parameters—in terms of camera trajectory consistency (CamMC). We achieved optimal results in directional accuracy (Flow Score) and 3D structure preservation (Depth Correlation), while achieving second-best performance in local stability (Warping Error) and generation quality.
>
> C2: Supplementary Video Evidence
>
> Due to the static nature of PDF files in the initial submission, we only provided frame-by-frame comparison maps. Adhering to double-blind requirements, we have provided an anonymous homepage (https://anonymous.4open.science/r/Anonymous-repository-of-Liewarper) showcasing dynamic results on the DAVIS 2017 and RealEstate10k datasets. We encourage you to review the page. It can be clearly observed that LieWarper maintains rigorous geometric rigidity and precise alignment with motion trajectories when handling complex composite movements of objects and cameras. These supplements intuitively demonstrate that our manifold geometric modeling ensures physical continuity across temporal spans.
>
> C3: Generation Quality Analysis
>
> Regarding the observation that Temp-Consistency and CLIP scores did not reach the absolute optimum in quantitative experiments, we provide a more detailed analysis:
>
> (1) LieWarper’s temporal consistency (0.9753) is lower than DiTFlow’s (0.9832) on the DAVIS2017 dataset. However, DiTFlow’s Flow-Score is only 0.3057, meaning the model tends to generate nearly static frames to avoid temporal deformation errors, as evidenced in the video results. In contrast, LieWarper maintains a high temporal performance of 0.9753 while achieving the highest Flow-Score (0.6907).
>
> (2) Geometric constraints impose a slight regularization on the diffusion model's space for semantic divergence, whereas methods lacking such constraints allow the diffusion model to freely degrade toward "CLIP-friendly" generation. Video evidence suggests the slight decline in CLIP is not a regression in quality, but rather a reasonable alignment cost for resisting geometric distortion.
>
> (3) In motion transfer tasks, trajectory alignment, 3D structure preservation, and stability are more critical metrics, in which LieWarping performs exceptionally.
>
> C4: Theoretical Boundaries of Optical Flow Dependency and Complex 3D Camera Motion
>
> (1) This study does not perform pixel-level warping on optical flow; instead, it calculates Lie algebra through analytical solving. This dimensionality reduction is inherently highly robust. Stress tests with Gaussian white noise injection in Appendix D prove that under extreme observation noise, our method's performance degradation (8.3%) is significantly lower than that of sub-optimal baseline methods (11.7%) that rely directly on dense optical flow warping.
>
> (2) Appendix C proves that forcibly re-projecting to the $SE(3)$ group without camera intrinsics leads to severe phase disruption due to the squared amplification of depth estimation errors. Therefore, using $Sim(2)$ as the base manifold—skillfully utilizing its scaling parameters to implicitly model visual depth zooming, complemented by residual fields to absorb local parallax—provides stability when handling complex lens movements. This has been fully validated by the new experiments on the RealEstate10k dataset.

---

> > ### Author Rebuttal · Reviewer_d4cj · 2026-04-03
> >
> > The authors have largely addressed my concerns through additional experiments.

---

> > > ### Author Response · Authors · 2026-04-06
> > >
> > > Thank you for your recognition and improvement of the score. Your suggestions have greatly enhanced our work.

---

### Official Review · Reviewer_E8TV · 2026-03-12

**Soundness:** 3
**Presentation:** 3
**Significance:** 3
**Originality:** 3
**Overall Recommendation:** 4
**Confidence:** 3

**Summary:**

LieWarper is a geometry-aware motion transfer framework that models motion as coordinate evolution on the Sim(2) Lie group manifold. The method decouples motion into (i) a global rigid reference frame (similarity transformation) solved analytically from HOSVD-filtered optical flow, and (ii) local non-rigid residuals that are transported and used to modulate diffusion features via flow-guided phase warping. An inference-time joint optimization refines alignment. Experiments on datasets such as DAVIS (plus ablations/user study) report improved trajectory fidelity and geometric/geometric stability versus training-free baselines while retaining generation quality.

**Compliance With Llm Reviewing Policy:**

Affirmed.

**Ethical Review Concerns:**

N/A (I am not flagging this paper for ethics review; see Limitations/ethical note above.)

**Final Justification:**

I maintain my recommendation of Weak Accept (4). The paper presents a well-motivated geometry-aware framework (LieWarper) that models video motion as coordinate evolution on the Sim(2) Lie group manifold, offering a principled alternative to Euclidean pixel-displacement approaches. The core strengths, rigorous geometric formulation, analytic solver with HOSVD filtering, and training-free RoPE-based phase modulation, remain solid contributions to the video motion transfer literature.

During the rebuttal, the authors addressed all five of my main concerns satisfactorily: (1) Flow robustness was demonstrated through noise injection experiments showing graceful degradation (8.3% vs. 11.7% for baselines) and cross-model consistency tests. (2) Multi-object and parallax handling was evidenced by new RealEstate10k experiments with diverse camera movements, where LieWarper achieved leading performance in trajectory consistency and depth correlation. (3) The continuity constraint via Lie algebra first-order difference regularization across sliding windows resolved the identity drift issue I flagged. (4) Orthogonal component ablations cleanly isolated the contribution of each module. (5) The authors committed to adding a misuse risk and mitigation section.

Remaining limitations, the fundamental Sim(2) planar approximation and reliance on optical flow quality, are honestly discussed and represent inherent scope boundaries rather than methodological flaws. The rebuttal reinforced my prior assessment that this is a technically sound paper with a meaningful contribution to geometry-aware video generation, though the scope of applicability (planar-dominant scenes) and the originality (creative combination of known tools rather than fundamentally new machinery) temper the impact. Overall, the strengths outweigh the weaknesses, and I believe the work merits acceptance.

**Key Questions For Authors:**

1) Flow dependence & failure cases: how robust is performance with poor optical flow (occlusion/blur), and is there a confidence estimate for the solved global motion? A convincing answer would increase my soundness score.
2) Multi-object / strong parallax: how does LieWarper behave when a single Sim(2) frame is insufficient (multiple independently moving objects, large depth variation)? More evidence would improve my significance rating.
3) Continuity constraints: can you incorporate continuity across windows (e.g., splines/regularization) without the identity-inconsistency issue? Demonstrating this would strengthen the approach.
4) Component ablations: can you clearly isolate contributions of (i) analytic solver, (ii) manifold constraint, (iii) phase modulation, and (iv) joint optimization? Cleaner ablations would improve clarity and originality claims.
5) Ethical considerations: how do you mitigate misuse for impersonation/deepfake-like content? A thoughtful discussion would improve my overall recommendation.

**Limitations:**

Limitations are discussed well (temporal resolution vs stability, Sim(2) planar approximation, and an SE(3) attempt/failure analysis). However, potential negative societal impacts (misuse for impersonation/deepfakes) are not addressed. I recommend adding a dedicated section on misuse risk, mitigation, and intended legitimate use cases.

**Strengths And Weaknesses:**

**Strengths**
- Strong motivation: clearly identifies perspective collapse/shearing as a failure mode of Euclidean motion control.
- Conceptually solid geometry-aware formulation (Sim(2) manifold evolution) with clear decoupling between global rigid motion and local residual dynamics.
- Specific analytic solver details (HOSVD filtering, structure tensor/projection vector formulation) and multiple evaluation metrics that cover trajectory fidelity, structural preservation, and perceptual quality.
- Useful ablations/analysis, including limitations: time-window vs global stability tradeoff and Sim(2) planar approximation vs SE(3) attempt and failure analysis.
- Practical benefits: training-free integration with pretrained video diffusion (e.g., WAN-2.1-1.3B) without retraining.

**Weaknesses**
- Heavy reliance on optical flow (and depth in some ablations) may limit robustness; it’s unclear how well it handles severe occlusion, motion blur, or in-the-wild video artifacts.
- The constant-velocity Lie algebra assumption is restrictive for erratic motion; the proposed sliding window fixes view-exit but introduces discontinuity and identity inconsistency.
- SE(3) attempt suggests depth noise is a major blocker; the paper’s solution ultimately remains Sim(2)-bound, which may struggle with strong parallax/multi-object scenes.
- Some derivations and the phase modulation details could be spelled out more explicitly for reproducibility.
- Ethical/societal implications (misuse for impersonation/deepfakes) are not discussed in detail and should be included.

---

> ### Author Rebuttal · Authors · 2026-03-31
>
> We sincerely thank you for your high appraisal of our modeling of Lie group manifold evolution and the design of the analytical solver, as well as for your affirmation regarding our ablation studies and discussion of limitations. In response to your questions, we provide the following clarifications.
>
> Visualized results are presented on our anonymous project page: https://anonymous.4open.science/r/Anonymous-repository-of-Liewarper.
>
> B1: Optical Flow Dependency
>
> (1) Experiments involving Gaussian white noise injection in Appendix D.1 demonstrate that at higher noise levels, the performance degradation of our method (8.3%) is significantly lower than that of the sub-optimal method "go-with-the-flow" (11.7%), which directly relies on dense optical flow warping.
>
> (2) Appendix D.2 proves that our method effectively generates coherent and consistent motion trajectories across various optical flow models, including traditional estimation methods and two categories of deep estimation models.
>
> B2: Multi-object/High Parallax
>
> (1) The ‘video4:dogs-jump’ case in the DAVIS 2017 dataset demonstrates the generation performance in multi-object scenarios.
>
> (2) The newly added RealEstate10k dataset contains extensive camera movements and parallax changes. For instance, visualized cases ‘video1’ and ‘video5’ feature large-scale camera zooming, while ‘video2’ and ‘video7’ involve camera translation and rotation. Our method continues to exhibit excellent motion fidelity and generation quality, proving the effectiveness of $Sim(2)$ manifold modeling.
>
> B3: Continuity Constraints
>
> While the method discussed in the original text optimizes different windows independently, we have since imposed smoothing regularization at the boundaries of adjacent windows by penalizing the first-order difference between the Lie algebras of adjacent windows ($\|\xi_{t+s} - \xi_t\|_2^2$). In the ‘video2:walking’ case from the DAVIS dataset, the identity drift previously caused by window switching has been eliminated. In the ‘video3:bmx-bumps’ case (which includes the subject moving out of frame, re-entering, and severe physical occlusion), the addition of this continuity constraint robustly maintained subject identity consistency. This evidence demonstrates that the inclusion of regularization effectively strengthens temporal continuity.
>
> B4: Component Ablation
>
> Here, we supplement this with an orthogonal ablation to isolate the independent contribution of each component.
>
> | Model Variant | FTD ↓ | Flow-Score ↑ | Depth-Corr ↑ | Warp-Err ↓ | CLIP Score ↑ | Temp-Con ↑ |
> | :--- | :--- | :--- | :--- | :--- | :--- | :--- |
> | LieWarper | 0.2381 | 0.6907 | 0.8418 | 19.3053 | 0.3154 | 0.9753 |
> | w/o Analytic Solver | 0.2501 | 0.6799 | 0.8303 | 24.4718 | 0.3137 | 0.9765 |
> | w/o Manifold Constraint | 0.2767 | 0.6771 | 0.7931 | 25.3382 | 0.3167 | 0.9754 |
> | w/o Phase Modulation | 0.5531 | 0.6071 | 0.7274 | 31.5739 | 0.3017 | 0.9706 |
> | w/o Joint Optimization | 0.4126 | 0.5426 | 0.7444 | 27.9324 | 0.3051 | 0.9745 |
>
> When the analytical solver degrades to stochastic initialization, the model searches blindly in a non-convex space and is prone to falling into local sub-optima, significantly harming trajectory consistency (FTD) and local stability (Warp-Err). Degrading the manifold constraint leads to a loss of global perspective constraints, causing a noticeable decline in trajectory consistency (FTD) and depth correlation (Depth-Corr). Completely removing phase modulation blocks the transmission of motion commands to the diffusion model, resulting in a failure of controlled generation. Eliminating joint optimization prevents the fitting of local residuals, leading to a comprehensive decline in all motion fidelity indicators.
>
> B5: Detailed Description and Ethical Considerations
>
> (1) In the analytical solution, we utilize the smoothed optical flow $\hat{\mathcal{F}}\_i$ and the Jacobian matrix $J(x\_i)$ to construct the structure tensor $\mathcal{A}=\sum J_i^T J_i$ and the projection vector $b=\sum J\_i^T \hat{\mathcal{F}}\_i$, deriving the Lie algebra closed-form solution $\xi\^\*=\mathcal{A}^{-1}b$. In phase modulation, the observed prior $P_{\mathcal{F}}^{(t)}$ and the non-rigid residual $\Delta P^{(t)}$, warped by the rigid skeleton $M\_t=\exp(t\cdot\xi^*)$, are superimposed to form the final coordinate $P'\_t$. We then utilize the operator $e^{i\langle\Theta, P'\_t\rangle}$ to directly modulate Query/Key features. Due to space constraints, we will add a more detailed description in the final version to ensure reproducibility.
>
> (2) In the Impact Statement section, we briefly mentioned the challenges high-fidelity generation poses to forgery detection. In the final version, we will add a section on "Misuse Risks and Mitigation Strategies." This will clearly define legitimate application scenarios, such as film visual effects and artistic creation, while embedding digital watermarks in the generated content to ensure traceability.

---

> > ### Author Rebuttal · Reviewer_E8TV · 2026-04-04
> >
> > My concerns have been addressed.

---

> > > ### Author Response · Authors · 2026-04-06
> > >
> > > Thank you sincerely for your recognition and help. Wish you all the best.

---

### Official Review · Reviewer_jyJb · 2026-03-15

**Soundness:** 3
**Presentation:** 3
**Significance:** 3
**Originality:** 2
**Overall Recommendation:** 4
**Confidence:** 3

**Summary:**

This paper presents LieWarper, a geometry-aware framework designed for video motion transfer. The authors identify a prevalent "coordinate blindness" in existing methods, which typically treat motion as unconstrained pixel displacements or linear phase shifts in Euclidean space—assumptions that often result in shearing artifacts or perspective collapse.To address this, LieWarper reformulates motion as coordinate evolution on the $Sim(2)$ Lie group manifold. The framework employs an analytic solver to extract global motion parameters from optical flow, stabilized by High-Order Singular Value Decomposition (HOSVD). These parameters then drive a flow-guided phase modulation mechanism that intervenes in the Rotary Positional Encodings (RoPE) of Diffusion Transformers (DiTs), allowing the model to adapt local non-rigid dynamics within a rigid global reference frame. The method is training-free and demonstrates superior geometric consistency and motion fidelity on the DAVIS dataset compared to contemporary baselines like RoPECraft and MotionClone

**Compliance With Llm Reviewing Policy:**

Affirmed.

**Final Justification:**

The video was great and solved my problem. I've decided to increase the rating.

**Key Questions For Authors:**

Computational Efficiency: The inference time is ~289s for 49 frames. Could you provide a breakdown of the time spent on RAFT flow extraction versus the HOSVD and the 5-step AdamW optimization?

While the paper provides extensive mathematical derivations and effectively achieves its results by modifying RoPE, it fails to provide any video comparison experiments. For a task that fundamentally requires demonstrating temporal consistency and motion fluidity, the absence of video evidence significantly undermines the credibility of the proposed method. Consequently, I recommend a Weak Reject.

**Limitations:**

Yes.

**Strengths And Weaknesses:**

# Strengths
Rigorous Geometric Foundation: Modeling motion via the exponential map $\exp(t \cdot \xi^*)$ ensures trajectories remain on the $Sim(2)$ manifold. This mathematically prevents "volume collapse" and shearing common in Euclidean linear interpolation.

Robust Spectral Denoising: The use of HOSVD to truncate high-frequency temporal modes effectively suppresses jitter and occlusion noise in raw flow. This ensures the analytic motion solver remains stable even with low signal-to-noise ratios.

Non-Invasive Latent Control: Modulating RoPE phases allows motion injection without altering feature values. This maintains the i.i.d. Gaussian distribution of pre-trained features, eliminating artifacts caused by out-of-distribution (OOD) shifts.

# Weakness

Missing Video Evidence: For a paper fundamentally centered on video motion transfer, the absence of a supplementary video is a significant drawback. While the authors provide static frame sequences in Figure 4 and Figure 11, these are insufficient to evaluate critical temporal artifacts such as flickering, texture swimming, or identity drifting.

Planar Approximation: $Sim(2)$ is limited to planar homography and cannot handle Z-dependent parallax. This leads to structural distortions in scenes with extreme depth variations, such as roller coasters.

---

> ### Author Rebuttal · Authors · 2026-03-31
>
> We appreciate your high evaluation of this research regarding its foundations in manifold geometry, spectral denoising robustness, and non-invasive latent control mechanisms. In response to your questions, we provide a detailed reply below.
>
> A1: Supplementary Video Evidence
>
> (1) Due to the constraints of static PDF documents in the initial submission, we were only able to display partial frame trajectory comparison maps. To better demonstrate the performance of our method, we have provided supplementary comparison videos between LieWarper and the baselines on an anonymous project page (https://anonymous.4open.science/r/Anonymous-repository-of-Liewarper). We cordially invite you to review this page. The supplementary video results clearly demonstrate that LieWarper maintains rigorous geometric rigidity and precise motion trajectory transfer even when handling complex composite movements of objects and camera lenses. This proves that modeling motion as coordinate manifold evolution not only preserves structural integrity in static single frames but also ensures temporal consistency and motion smoothness across the sequence, while maintaining high fidelity to the text prompt.
>
> (2) Furthermore, incorporating suggestions from other reviewers, we have included two additional comparison methods (MotionCtrl [1] and CameraCtrl [2]) and extended our experiments on the RealEstate10k dataset. The visualization results are also available for viewing on the anonymous project page. The experimental results indicate that our method outperforms the fine-tuning-based MotionCtrl and CameraCtrl—which require explicit camera trajectory parameters—in terms of 3D structure preservation and motion fidelity. This further demonstrates the significant potential of geometric priors in guiding pre-trained diffusion models.
>
> A2: Computational Efficiency and Time Breakdown
>
> As indicated in the main text, the average processing time per video on the DAVIS dataset using a single A100 is 289.05 seconds. Specifically, the optical flow extraction stage based on the RAFT model takes approximately 1.91 seconds. The analytical solving process for HOSVD filtering and global similarity transformation, benefiting from the design of pure mathematical closed-form solutions, takes only 0.18 seconds. During the inference stage, the iterative update involving 5 steps of AdamW takes approximately 200 seconds, with the remaining time consisting of the standard inference overhead for forward denoising from the base backbone model (Wan-2.1-1.3B).
>
> A3: Regarding the Limitations of $Sim(2)$ Planar Approximation
>
> (1) As we explored in Appendix C, although the full 6-DOF $SE(3)$ group can theoretically handle 3D parallax, practical scenarios lacking ground-truth camera intrinsics must rely on external monocular depth estimation for coordinate re-projection. However, the error gain in depth estimation is inversely proportional to the square of the depth. Such amplified high-frequency coordinate jitter directly disrupts the strict requirements of rotary positional embeddings for phase spectral continuity, leading to the collapse of the self-attention mechanism.
>
> (2) The newly added RealEstate10k dataset contains extensive camera movements, including panning, tilting, zooming, and rotating, with significant changes in scene depth; nonetheless, our method still exhibits excellent trajectory consistency and generation quality. Therefore, for monocular video generation, adopting $Sim(2)$ constraints represents the optimal trade-off between geometric preservation and generative robustness.
>
> [1] Wang, Zhouxia, et al. "Motionctrl: A unified and flexible motion controller for video generation." ACM SIGGRAPH 2024.
>
> [2] He H, Xu Y, Guo Y, et al. "Cameractrl: Enabling camera control for video diffusion models" ICLR 2025.

---

> > ### Author Rebuttal · Reviewer_jyJb · 2026-04-05
> >
> > The video was great and solved my problem. I've decided to increase the rating

---

> > > ### Author Response · Authors · 2026-04-06
> > >
> > > Thanks for your acknowledgment of our work and for raising the score. Thanks again for your time and effort in reviewing our paper.

---

### Decision · Program_Chairs · 2026-04-30

**Decision:**

Accept (regular)

**Comment:**

The submission presents LieWarper, a geometry-awaremotion transfer framework that models motion as coordinate evolution on the Sim(2) Lie group manifold. It combines an analytic motion solver with flow-guided phase modulation to preserve global geometry while handling local non-rigid dynamics. All reviewers initially recommended weak accept with moderate confidence, due to its strong mathematical formulation and practical performance, but also raised concerns about limited evaluation, lack of video evidence, dependence on optical-flow performance, and the planar Sim(2) limitation. The rebuttal address the main concerns by adding supplementary videos, experiments on RealEstate10k, more ablations and runtime breakdowns. All reviewers explicitly acknowledged that their concerns were resolved and indicated increasing scores or keeping positive ratings. Hence, a weak accept recommendation is suggested.